# Higher native Peruvian genetic ancestry proportion is associated with tuberculosis progression risk

## Graphical abstract

## Authors

Samira Asgari, Yang Luo,
Chuan-Chin Huang, ..., D. Branch Moody,
Megan B. Murray, Soumya Raychaudhuri

## Correspondence

megan_murray@hms.harvard.edu
(M.B.M.),
soumya@broadinstitute.org (S.R.)

## In brief

Our understanding of how genetic differences among human populations may affect susceptibility to infectious diseases is very limited. Asgari et al. show that the proportion of native genetic ancestry in contemporary Peruvians affects the risk of progression from latent to active tuberculosis even after accounting for differences in socio-demographic factors.

## Highlights

- Genotyping 3,425 Peruvians with active TB and controls with latent tuberculosis

- Longitudinal, household-contact study design

- Higher native Peruvian genetic ancestry increases tuberculosis progression risk

- Socio-demographic factors do not explain this association

Asgari et al., 2022, Cell Genomics 2, 100151
July 13, 2022 © 2022 The Author(s).

CellPress

## Article

# Higher native Peruvian genetic ancestry proportion is associated with tuberculosis progression risk

Samira Asgari,[1,2,3,4,5,13] Yang Luo,[1,2,3,4,5] Chuan-Chin Huang,[6,8] Zibiao Zhang,[6,7,8] Roger Calderon,[10] Judith Jimenez,[10] Rosa Yataco,[10] Carmen Contreras,[10] Jerome T. Galea,[6,11] Leonid Lecca,[6,10] David Jones,[6,9] D. Branch Moody,[2] Megan B. Murray,[6,7,8,*] and Soumya Raychaudhuri[1,2,3,4,5,12,14,*]

[1]Center for Data Sciences, Brigham and Women's Hospital, Harvard Medical School, Boston, MA, USA
[2]Division of Rheumatology, Inflammation, and Immunity, Brigham and Women's Hospital, Harvard Medical School, Boston, MA, USA
[3]Division of Genetics, Brigham and Women's Hospital, Harvard Medical School, Boston, MA, USA
[4]Broad Institute of MIT and Harvard, Cambridge, MA, USA
[5]Department of Biomedical Informatics, Harvard Medical School, Boston, MA, USA
[6]Department of Global Health and Social Medicine, Harvard Medical School, 641 Huntington Avenue, Boston, MA 02115, USA
[7]Department of Epidemiology, Harvard School of Public Health, Boston, MA, USA
[8]Division of Global Health Equity, Brigham and Women's Hospital, Boston, MA, USA
[9]Department of the History of Science, Harvard University, Cambridge, MA, USA
[10]Socios En Salud, Lima, Perú
[11]School of Social Work, University of South Florida, Tampa, FL, USA
[12]Centre for Genetics and Genomics Versus Arthritis, Manchester Academic Health Science Centre, University of Manchester, Manchester, UK
[13]Institute for Genomic Health, Icahn School of Medicine at Mount Sinai, New York, USA
[14]Lead contact
*Correspondence: megan_murray@hms.harvard.edu (M.B.M.), soumya@broadinstitute.org (S.R.)

## SUMMARY

We investigated whether ancestry-specific genetic factors affect tuberculosis (TB) progression risk in a cohort of admixed Peruvians. We genotyped 2,105 patients with TB and 1,320 household contacts (HHCs) who were infected with *Mycobacterium tuberculosis* (*M. tb*) but did not develop TB and inferred each individual's proportion of native Peruvian genetic ancestry. Our HHC study design and our data on potential confounders allowed us to demonstrate increased risk independent of socioeconomic factors. A 10% increase in individual-level native Peruvian genetic ancestry proportion corresponded to a 25% increased TB progression risk. This corresponds to a 3-fold increased risk for individuals in the highest decile of native Peruvian genetic ancestry versus the lowest decile, making native Peruvian genetic ancestry comparable in effect to clinical factors such as diabetes. Our results suggest that genetic ancestry is a major contributor to TB progression risk and highlight the value of including diverse populations in host genetic studies.

## INTRODUCTION

Tuberculosis (TB), caused by *Mycobacterium tuberculosis* (*M. tb*), is the leading cause of death from an infectious disease worldwide.[1,2] Similar to other infectious diseases, the development of TB after *M. tb* infection is determined in part by human genetic factors.[3] Previous twin studies have shown that under comparable environmental and social conditions, TB concordance is higher in monozygotic twins than in dizygotic twins.[3] Similarly, human genomics studies of TB have identified a number of variants that are associated with TB risk.[4–7] However, there is little concordance between known TB susceptibility loci in different populations,[3] suggesting that the risk alleles driving TB risk in different populations may be heterogeneous.

Because pathogens are a major selective force in shaping our genome,[8] it is reasonable to think that the high historical prevalence of TB in Europe over the past 2,000 years may have led to reduced frequencies of risk alleles in the European population. Indeed, a recent study showed that the negative selection exerted by the high burden of TB is likely to explain the sharp drop in the frequency of rs34536443, a missense variant in *TYK2* that confers TB risk, after the Bronze age (2,500 years ago).[9] Similarly, a previous study of TB risk among admixed South Africans showed that European genetic ancestry protects against TB disease.[10] It is thus plausible that genetic ancestry contributes to differences in the incidence of TB across populations. However, quantifying the contribution of ancestry-specific genetic factors to TB risk can be challenging because genetic

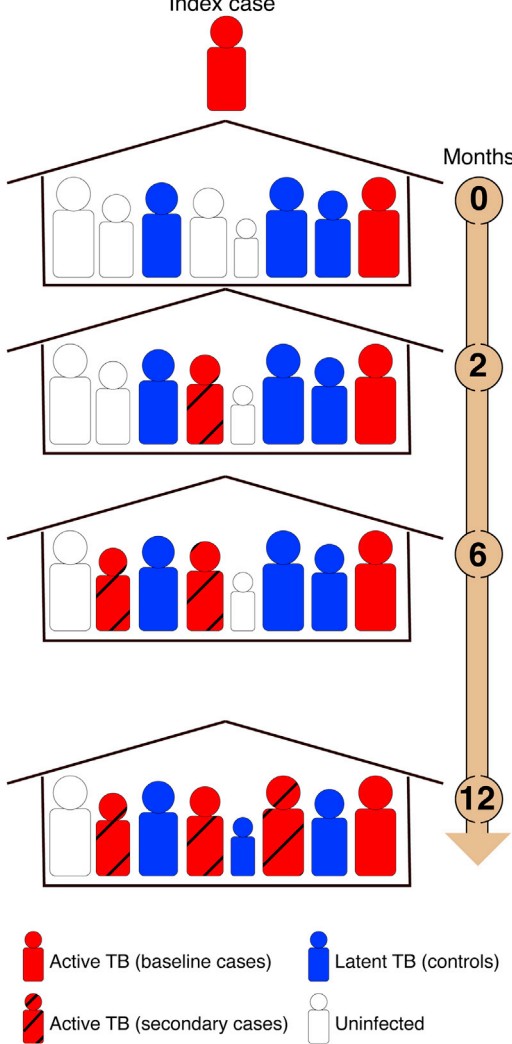

Index case

Months

Active TB (baseline cases)

Active TB (secondary cases)

Latent TB (controls)

Uninfected

**Figure 1. Our household-contact study design**
We recruited patients in a large catchment area that included 20 urban districts and ~3.3 million residents. Within 14 days of enrollment of index cases, we contacted their household contacts (HHCs). HHCs with pulmonary TB were recruited as cases (baseline cases). HHCs that were TST positive but did not have active TB were recruited as controls. All individuals were followed up with for 1 year, and all HHCs were evaluated for signs and symptoms of pulmonary and extra-pulmonary TB disease at 2, 6, and 12 months after enrollment and were recruited as cases if they developed active TB during follow up (secondary cases). HHCs that remained or became TST positive but did not develop active TB were recruited as controls. The final cohort included 2,105 TB cases and 1,320 TST-positive HHCs.

ancestry can track with non-genetic sociodemographic TB risk factors, such as smoking and under-nutrition.[2]

Here, we aim to understand the role of ancestry-specific genetic factors that affect TB progression risk independently of socioeconomic and environmental factors in a cohort of Peruvian individuals. Peru has one of the highest TB incidences in South America.[11] The genetic makeup of contemporary Peruvians is shaped by extensive admixture between native residents of Peru and the Europeans, Africans, and Asians that have arrived

in Peru since the 16th century.[12] We recruited patients with TB and *M. tb*-infected household contacts (HHCs) in whom we ascertained infection status by tuberculin skin testing (TST). We specifically picked controls in this way to make sure they were exposed and infected and to focus specifically on TB progression risk. We also ascertained sociodemographic and known TB clinical risk factors in all participants. We then used genotype data to quantify the genetic diversity in our cohort and to estimate the proportion of native Peruvian genetic ancestry (i.e., the indigenous genetic ancestry component of the genome of contemporary Peruvians) in each individual. Finally, we tested the association between genetic ancestry and TB progression risk after accounting for potential confounding effects.

## RESULTS

### Study design and case-control definition
We conducted a longitudinal, HHC genetic study of pulmonary TB in Lima, Peru (STAR Methods; Figures S1 and S2). All cases (n = 2,105) had confirmed active TB. Within 14 days of enrollment of index TB cases (i.e., the first TB patient in each household), we screened their HHCs for signs and symptoms of active TB as well as for latent TB as measured by a TST. These tests were repeated at 2, 6, and 12 months (STAR Methods; Figure 1). We refer to HHCs who were identified as having TB within 14 days of enrollment of index TB cases as "baseline cases" and to HHCs who were diagnosed with TB after this period until the end of the 12 months follow up as "secondary" or "secondary clustered cases" (see STAR Methods for details). Controls (n = 1,320) are HHCs of index cases who were TST positive but who did not develop TB during 12 months of active follow up (STAR Methods). In addition to individuals' TB status, we also collected extensive information on sociodemographic risk factors for TB (STAR Methods; Table 1).

### Global ancestry inference
We quantified the global genetic ancestry for each individual in our cohort, assuming four ancestral populations (K = 4) based on Peru's population history[12] (Figure S3). These four populations corresponded to native Peruvian, European, West African, and East Asian genetic ancestry with average proportions 0.80 (standard deviation [SD] = 0.15), 0.16 (0.11), 0.03 (0.07), and 0.01 (0.03), respectively (Table 1; Figure 2A; Table S1; Figure S4). These proportions were consistent with previous genetic studies of Peruvians.[12] Increasing the number of clusters revealed finer substructures within each of the four main ancestral clusters (Figure S5; Table S2).

### Correlation between self-reported race and genetic ancestry
Self-reported race or ethnicity is frequently used in epidemiological or medical studies to account for an individual's background. However, self-reported race/ethnicity can be a poor proxy for genetic ancestry in admixed populations.[13–15] In our cohort, the majority of participants self-identify their race as "American Indian + White" and their ethnicity as "Latino" (74% and 99% respectively; Table 2). Genetic ancestry proportions differ significantly between self-reported race categories (ANOVA p < 10^-30 for all four tested

**Table 1. Cohort's demographic information**

| | Mean (SD) | | | | p value |
|---|---|---|---|---|---|
| **Native Peruvian genetic ancestry** | | | | | |
| NAT tertile 1 | 0.64 (0.13) | | | | $<2.2 \times 10^{-308}$ |
| NAT tertile 2 | 0.84 (0.03) | | | | |
| NAT tertile 3 | 0.94 (0.03) | | | | |
| **European genetic ancestry** | | | | | |
| NAT tertile 1 | 0.26 (0.10) | | | | $<2.2 \times 10^{-308}$ |
| NAT tertile 2 | 0.14 (0.03) | | | | |
| NAT tertile 3 | 0.06 (0.03) | | | | |
| **West African genetic ancestry** | | | | | |
| NAT tertile 1 | 0.08 (0.10) | | | | $3.6 \times 10^{-185}$ |
| NAT tertile 2 | 0.015 (0.02) | | | | |
| NAT tertile 3 | 0.005 (0.006) | | | | |
| **East Asian genetic ancestry** | | | | | |
| NAT tertile 1 | 0.02 (0.05) | | | | $1.4 \times 10^{-59}$ |
| NAT tertile 2 | 0.007 (0.01) | | | | |
| NAT tertile 3 | 0.003 (0.006) | | | | |
| **Age** | | | | | |
| NAT tertile 1 | 33.26 (15.59) | | | | $2.0 \times 10^{-8}$ |
| NAT tertile 2 | 29.02 (13.12) | | | | |
| NAT tertile 3 | 34.90 (17.05) | | | | |
| **TB status** | **control (%)** | **case (%)** | | | |
| NAT tertile 1 | 0.16 | 0.17 | | | $2.3 \times 10^{-16}$ |
| NAT tertile 2 | 0.12 | 0.22 | | | |
| NAT tertile 3 | 0.11 | 0.23 | | | |
| **Sex** | **female (%)** | **male (%)** | | | |
| NAT tertile 1 | 0.13 | 0.20 | | | $4.7 \times 10^{-6}$ |
| NAT tertile 2 | 0.14 | 0.19 | | | |
| NAT tertile 3 | 0.16 | 0.17 | | | |
| **Smoking status** | **heavy (%)** | **light (%)** | **non-smoker (%)** | **NA (%)** | |
| NAT tertile 1 | 0.02 | 0.02 | 0.29 | 0.008 | $2.2 \times 10^{-26}$ |
| NAT tertile 2 | 0.004 | 0.01 | 0.31 | 0.007 | |
| NAT tertile 3 | 0.002 | 0.004 | 0.32 | 0.005 | |
| **Drinking status** | **heavy (%)** | **light (%)** | **non-drinker (%)** | **NA (%)** | |
| NAT tertile 1 | 0.04 | 0.11 | 0.17 | 0.02 | $1.5 \times 10^{-12}$ |
| NAT tertile 2 | 0.03 | 0.09 | 0.2 | 0.01 | |
| NAT tertile 3 | 0.02 | 0.09 | 0.22 | 0.01 | |
| **Body mass index** | **normal (%)** | **overweight (%)** | **underweight (%)** | **NA (%)** | |
| NAT tertile 1 | 0.007 | 0.04 | 0.28 | 0.004 | 0.12 |
| NAT tertile 2 | 0.01 | 0.04 | 0.28 | 0.004 | |
| NAT tertile 3 | 0.007 | 0.03 | 0.29 | 0.004 | |
| **Previous TB** | **No (%)** | **Yes (%)** | | | |
| NAT tertile 1 | 0.31 | 0.03 | | | 0.09 |
| NAT tertile 2 | 0.30 | 0.04 | | | |
| NAT tertile 3 | 0.30 | 0.04 | | | |

*(Continued on next page)*

**Table 1.** *Continued*

| | Mean (SD) | | | | p value |
|---|---|---|---|---|---|
| **Education** | **above high school (%)** | **below high school (%)** | | | |
| NAT tertile 1 | 0.1 | 0.23 | | | $4.7 \times 10^{-7}$ |
| NAT tertile 2 | 0.09 | 0.24 | | | |
| NAT tertile 3 | 0.07 | 0.26 | | | |
| **BCG vaccination** | **no (%)** | **yes (%)** | | | |
| NAT tertile 1 | 0.03 | 0.3 | | | $5.6 \times 10^{-04}$ |
| NAT tertile 2 | 0.04 | 0.29 | | | |
| NAT tertile 3 | 0.05 | 0.28 | | | |
| **Socioeconomic group** | **low (%)** | **middle (%)** | **high (%)** | **NA (%)** | |
| NAT tertile 1 | 0.10 | 0.13 | 0.08 | 0.01 | 0.79 |
| NAT tertile 2 | 0.11 | 0.13 | 0.08 | 0.01 | |
| NAT tertile 3 | 0.11 | 0.14 | 0.08 | 0.01 | |

The cohort includes 2,105 individuals with active TB (cases) and 1,320 infected HHCs (controls). We divided the cohort into tertiles based on native Peruvian genetic ancestry and tested the association of these tertiles with each covariate individually. Two-sided p values are calculated using the ANOVA for quantitative variables and using the chi-square test for categorical variables. For quantitative variables, mean (SD), and for categorical variables, percentages are shown. NA, not available. The socioeconomic group is a household-level variable; all other variables are individual-level variables. Numbers are rounded to two or three decimal places.

genetic ancestries; Figure 3). For example, individuals who self-identified as "Black" had a higher proportion of West African genetic ancestry than the average of all other categories (mean [SD] = 0.25 [0.23] versus 0.3 [0.6]; Figure 3). Nonetheless, native Peruvian ancestry was the dominant genetic ancestry in all categories of self-reported race (Figure 3). Similarly, 18% of individuals with high (>0.9) proportion of native Peruvian genetic ancestry (n = 985) self-identified as "American Indian" compared with only 7% of individuals with low (<0.5) proportion of native Peruvian genetic ancestry (n = 140; Tables S3 and S4). In all tertiles of native Peruvian ancestry, the majority of individuals self-reported as "American Indian + White" followed by "American Indian" (Table S5). Altogether, these results suggest that self-reported race and genetic ancestry are correlated; however, individuals who self-report the same race can have drastically different levels of genetic ancestry proportions.

We then tested whether self-reported race is associated with TB progression risk. No category of self-reported race was significantly associated with TB status, suggesting that in our cohort, self-reported race is not a risk factor for TB progression (Table S6).

### Association between genetic ancestry and TB progression risk

To examine the relationship between genetic ancestry and TB progression risk in Peruvians, we applied logistic regression to test the effect of the estimated fraction of native Peruvian, European, West African, and East Asian genetic ancestries on case-control status after adjusting for age, sex, and socioeconomic status. Additionally, we included a random household effect to account for environmental factors and a genetic relatedness matrix to account for cryptic relatedness between individuals. We observed a significant association between increased native Peruvian genetic ancestry and TB progression risk (odds ratio per 0.1 increase in native Peruvian genetic ancestry proportion [$OR_{NAT0.1}$] = 1.25, 95% confidence interval [CI] = 1.18–1.33, p =

$1.1 \times 10^{-13}$; Table 3), and European, West African, and East Asian genetic ancestries were associated with reduced TB progression risk (Table 3). Adjusting for self-reported race (Table S7) or removing 430 related individuals (kinship coefficient $\geq$ 0.125) did not change these results (Figure S6; Table S8). Similarly, stratifying by sex did not change our results (Figure S7; Table S9).

Next, to test whether these associations were independent of each other, we performed conditional analyses between ancestries. Native Peruvian genetic ancestry remained significantly associated with increased TB progression risk conditioned on the other ancestries, but the other ancestries showed no association with TB progression risk after conditioning on native Peruvian genetic ancestry (Table 3).

In our cohort, native Peruvian is the main genetic ancestry and the only one that is associated with an increased TB progression risk relative to other ancestry components. We observed a significantly higher level of native Peruvian genetic ancestry in cases compared with the infected HHCs (0.82 [SD = 0.13] and 0.78 [0.17], t test p = $8.8 \times 10^{-19}$; Figure 2B) and a higher probability of being a case with an increasing proportion of native Peruvian genetic ancestry. Individuals with the highest level of native Peruvian genetic ancestry (top decile, average native Peruvian genetic ancestry proportion = 0.97 [0.01], n = 232 cases, 110 controls) were three times more likely to progress to active TB (OR = 2.90, 95% CI = 1.99–4.26, p = $2.8 \times 10^{-8}$; Figure 2C) compared with the individuals with the lowest level of native Peruvian genetic ancestry (bottom decile, average native Peruvian genetic ancestry proportion = 0.48 [0.13], n = 149 cases, 194 controls). Assuming a larger number of ancestral clusters did not substantively change the association between native Peruvian genetic ancestry and TB progression risk (Table S10).

As a sensitivity analysis and to rule out the effect of individual-level non-genetic confounders, we added West African and East Asian genetic ancestry proportions, BMI, education level, and BCG vaccination, smoking, alcohol use, and previous TB status to our model. Including these additional covariates did not

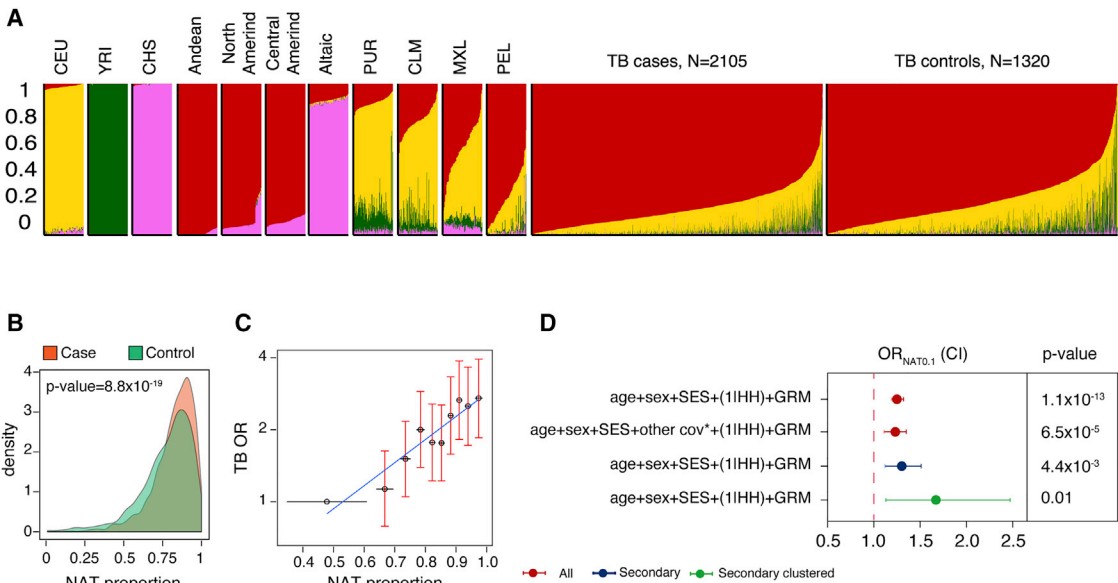

**Figure 2. Native Peruvian genetic ancestry is associated with TB progression risk**

(A) ADMIXTURE analysis results using K = 4 clusters. The average proportions of native Peruvian (red), European (yellow), West African (green), and East Asian (pink) genetic ancestry were 0.8 (standard deviation [SD] = 0.15), 0.16 (0.11), 0.03 (0.07), and 0.01 (0.03), respectively. X axis: individuals (axis ticks not shown). Y axis: genomic proportion. Displayed populations from left to right: Utah residents with Northern and Western European ancestry (CEU from the 1000 Genomes Project), Yoruba (YRI from the 1000 Genomes Project), Southern Han Chinese (CHS from the 1000 Genomes Project), Andean (from Reich et al.'s study), Northern American-Indians (North Amerind, from Reich et al.'s study), Central American-Indians (Central Amerind, from Reich et al.'s study), Altaic (Siberians speaking Altaic languages, from Reich et al.'s study), Puerto Ricans from Puerto Rico (PUR from the 1000 Genomes Project), Colombians from Medellin, Colombia (CLM from the 1000 Genomes Project), Mexican Ancestry from Los Angeles USA (MXL from the 1000 Genomes Project), Peruvians from Lima, Peru (PEL from the 1000 Genomes Project), TB cases form this study (n = 2,106), and TB controls from this study (n = 1,320).

(B) Probability density distribution of native Peruvian genetic ancestry proportion in TB cases and controls. TB cases have a higher proportion of native Peruvian genetic ancestry than infected HHCs (two-sided t test $p = 8.8 \times 10^{-19}$). Y axis: density. X axis: native Peruvian genetic ancestry (NAT) proportion.

(C) Individuals in the higher native Peruvian genetic ancestry decile have a higher TB risk. Individuals with the highest level of native Peruvian genetic ancestry (top decile, average native Peruvian genetic ancestry proportion = 0.97 [0.01], n = 232 cases, 110 controls) are three times more likely to progress to active TB (OR = 2.90, 95% CI = 1.99–4.26, $p = 2.8 \times 10^{-8}$) compared with the individuals in the bottom decile (average native Peruvian genetic ancestry proportion = 0.48 [0.13], n = 149 cases, 194 controls). X axis: average native Peruvian genetic ancestry proportion for each decile; error bars show standard error. Y axis: TB odds ratio (OR) after correction for age, sex, socioeconomic status, household, and genetic relatedness. ORs are shown relative to the first decile. Error bars show 95% CI. Each decile includes 398 individuals.

(D) Native Peruvian genetic ancestry remained significantly associated with TB progression risk after controlling for eight additional individual covariates including West African and East Asian genetic ancestry proportion, smoking, drinking, previous TB status, education level, BMI, and BCG vaccination. Similarly, native Peruvian genetic ancestry remained significantly associated with TB progression risk when we restricted our cohort to secondary or secondary clustered cases and their HHCs. Circles show OR for 0.1 increase in native Peruvian genetic ancestry; error bars show 95% CI. NAT, native Peruvian genetic ancestry.

change the observed association between native Peruvian genetic ancestry and TB progression risk ($OR_{NAT0.1}$ = 1.23 [1.11–1.35], $p = 6.5 \times 10^{-5}$; Figure 2D; Table S11).

Collectively, these results suggest that native Peruvian genetic ancestry is associated with increased TB progression risk independently of other genetic ancestries or non-genetic factors that can track with genetic ancestry such as sociodemographic or known clinical TB risk factors. However, these results do not rule out the possibility of this association being the result of other non-genetic confounders related to phenotypic heterogeneity, exposure, or transmission. We thus performed a series of statistical analyses to account for these potential confounders.

### Accounting for phenotypic heterogeneity
To test if phenotypic heterogeneity in our cohort could explain our results, we restricted the analysis to microbiologically confirmed TB cases (n = 2,043) and HHCs who were TST positive

at baseline and did not develop active TB during the 1 year follow up (n = 950). These analyses resulted in an OR similar to the larger cohort ($OR_{NAT0.1}$ = 1.28 [1.20–1.37], $p = 5.6 \times 10^{-14}$; Table S12). We also considered that cases and controls who were from the same household may share a more similar ancestry profile compared with average, which may bias our results. For this, we tested the association of native Peruvian genetic ancestry with TB progression risk using half of the cases (n = 791) and the same number of controls who were not from the same household as cases and after correction for age, sex, socioeconomic status, and genetic relatedness. This analysis had a similar result to the analysis performed using the whole cohort ($OR_{NAT0.1}$ = 1.19 [1.10–1.28], $p = 1.0 \times 10^{-12}$).

### Accounting for transmission and exposure
While index cases might have acquired TB in the community, secondary cases are more likely to result from within household

| Table 2. Self-reported race and ethnicity in our cohort (n = 3,425) | |
|---|---|
| Self-reported race | Count |
| American Indian + White | 2,538 |
| American Indian | 515 |
| White | 289 |
| Black | 42 |
| American Indian + Black | 15 |
| Asian | 6 |
| Black + White | 6 |
| American Indian + White + Asian | 1 |
| NA | 13 |
| Self-reported ethnicity | Count |
| Latino | 3,410 |
| Not Latino | 9 |
| NA | 6 |

transmission. To account for potential differences in exposure and transmission between index cases and HHCs, we tested the association between native Peruvian genetic ancestry and TB progression in secondary cases (n = 213) and controls from the same households (n = 214) and after adjusting for age, sex, socioeconomic status, household, and genetic relatedness, as we did in our primary analysis. We observed an OR similar to the one observed for the whole cohort ($OR_{NAT0.1} = 1.30$ [1.12–1.51], p = $4.4 \times 10^{-3}$; Figure 2D; Table S12).

We further restricted the cohort to secondary clustered TB cases to ensure that their TB disease was the result of within household transmission or infection from circulating *M. tb* strains rather than reactivation of an old infection. While this analysis had a much smaller sample size, the results were consistent with our previous analyses (n = 58 TB cases and 48 HHCs, $OR_{NAT0.1} = 1.67$ [1.13–2.47], p = $1.0 \times 10^{-2}$; Figure 2D; Table S12). Native Peruvian genetic ancestry proportion was similar between baseline and secondary cases from the same households, which is consistent with the conclusion that differences in native Peruvian genetic ancestry proportion are not associated with differences in exposure (n = 135 baseline and 213 secondary cases, $OR_{NAT0.1} = 1.02$ [0.86–1.22], p = 0.28; Table S11).

### Admixture mapping

We performed local ancestry inference followed by admixture mapping to look for specific genomic regions that might explain the association between native Peruvian genetic ancestry and TB progression risk (n = 889,203 markers following imputation, quality control [QC], and pruning). No locus passed the genome-wide significance threshold (p < $4.3 \times 10^{-6}$). However, we observed suggestive evidence of association at 5p23.2 (OR = 1.34 [1.17–1.53], p = $2.9 \times 10^{-5}$; Figure S8). When we restricted the analysis to cases with microbiologically confirmed TB (n = 2,043) and HHCs who were TST positive at baseline and did not progress to active TB over the 1 year follow up (n = 950), the signal on 5p23.2 got stronger (OR = 1.39 [1.20–1.61], p = $1.5 \times 10^{-5}$; Figure S8) and closer to the genome-wide significance threshold set using permutation for this analysis

(p < $1.0 \times 10^{-5}$). This locus overlaps a 100 Mb region on chromosome 5 (125855350-125963352), which includes the coding sequence of *ALDH4A1* and 69 variants that were nominally associated with TB progression risk in our cohort (Table S13).[4–7] However, understanding whether any of these variants or other variants in this locus might explain the observed admixture mapping signal requires further investigation.

### DISCUSSION

Our results suggest that relative to other tested genetic ancestries, native Peruvian genetic ancestry is associated with TB progression risk independently of population structure, the sociodemographic factors that we tested here, and factors related to exposure or transmission. In our cohort, individuals with the highest proportion of native Peruvian genetic ancestry are three times more likely to progress to active TB compared with individuals with the lowest level of native Peruvian genetic ancestry. To compare, this effect is similar to the reported effect of diabetes on TB risk based on previous cohort studies.[16]

Compared with native Peruvian genetic ancestry, European and West African genetic ancestries were associated with reduced TB progression risk. This protective effect can be the result of the long, shared history of these populations with *M. tb*,[17] which could have led to selective pressures that have mitigated TB genetic risk if such pressures were not present in pre-colonization Peru.[18] However, we want to emphasize that the effect of native Peruvian ancestry on TB progression risk is relative to other ancestries that we tested here and cannot be causally detangled from the effect of other genetic ancestries.

While our results show a strong genome-wide signal for the effect of native Peruvian genetic ancestry on TB risk, we did not identify any single locus that can explain this effect, suggesting that it is driven by a polygenic architecture with many variants exerting modest impact. This conclusion is in line with our previous genome-wide association study (GWAS) of TB progression in the current cohort where we showed SNP heritability ($h^2_g$) of TB progression in Peruvians to be 21.2% yet found only one genome-wide significant locus (3q23) associated with TB progression risk.[7] We did not identify any association in 3q23 locus in our admixture mapping analysis; however, GWAS and admixture mapping results can be complementary and do not necessarily always point to the same risk loci.[19,20] In addition to the polygenic structure or TB progression, our power to detect specific TB progression risk loci through admixture mapping may have also been affected by the lower accuracy of local ancestry inference in multi-way admixture scenarios compared with two-way admixtures.[10] Altogether, these results suggest that conclusively identifying TB progression risk loci requires larger studies with greater statistical power.

To our knowledge, this study is the first large-scale genetic study to look at the effect of indigenous ancestry on TB or TB progression risk in South or Latin American populations. However, the role of indigenous ancestry in the apparently increased burden of TB among native populations of America has been extensively debated for over 200 years.[21,22] These debates are rooted in epidemiological studies showing a high TB burden and mortality rates in post-contact indigenous

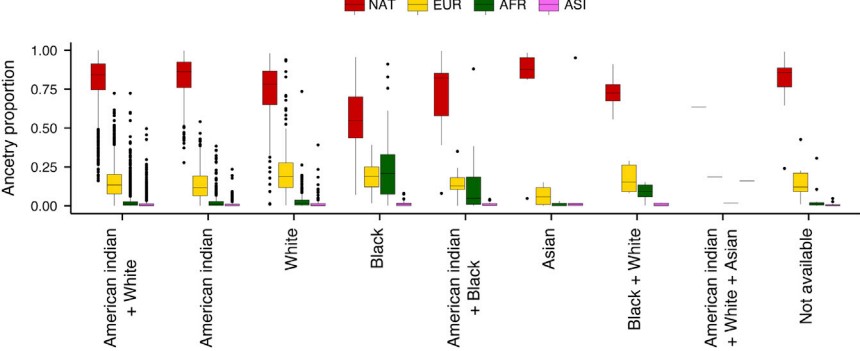

**Figure 3. Genetic ancestry proportions among different self-reported race categories**

Native Peruvian (NAT) is the dominant genetic ancestry in all categories of self-reported race. The proportion of NAT, European (EUR), West African (AFR), and East Asian (ASI) genetic ancestries was significantly different between self-reported race categories (ANOVA, degrees of freedom = 7, two-sided $p = 1.7 \times 10^{-48}$, $1.6 \times 10^{-32}$, $1.2 \times 10^{-118}$, and $8.2 \times 10^{-32}$, respectively). Box plots show median and interquartile range (IQR), and whiskers show 1.5× IQR.

Americans compared with Europeans.[21,22] More recent evidence for the role of genetic factors related to indigeneity and TB risk comes from candidate gene[23] or family[24] studies. However, these debates remain inconclusive mainly due to the challenges associated with separating population-specific genetic risk factors from non-genetic risk factors that track with genetic ancestry.[21,22]

Our study is different from these previous genetic studies in three ways. First, our study uses genetic data to quantitatively assign genetic ancestry, whereas previous studies used self-reported ancestry, which is often a poor proxy for genetic ancestry in admixed populations[13,15] and thus can lead to misclassification of participants. Second, we carefully phenotyped all individuals and ascertained infection status using TST to ensure that all individuals were exposed to *M. tb.* This is an important distinction as different genetic factors might underlie different stages of the disease (e.g., infection versus progression upon infection).[7] Third, our HHC, longitudinal study design allowed us to ensure that all controls were exposed to *M. tb* and to rigorously account for potential non-genetic factors that track with genetic ancestry.

In addition to its relevance for better understanding the genetic architecture of TB progression, our study provides a framework for similar future studies where it is important to account for environmental and socioeconomic factors to identify genetic factors that affect disease outcomes. Our results also highlight that differences in infectious disease burden among different populations cannot be solely attributed to variations in sociodemographic factors and can be partially due to genetic differences. Our results also highlight that differences in infectious disease burden among different populations cannot be solely attributed to variations in sociodemographic factors and can be partially due to genetic differences. Currently, the majority of human genomics studies of complex traits are done in populations of European ancestry.[25] However, with the increasing clinical applications of complex trait genomics data,[26] this European bias can lead to increased health disparities.[27] Our results underline the importance of conducting large-scale human genomics studies in diverse populations in order to get a better understanding of population-specific genetic risk for TB and other complex diseases, to get a comprehensive picture of the genotype-phenotype relationship, and to enable all human populations to benefit from the results of human genomics research.

## Limitations of the study

One caveat of our study is that we have not tested for all possible non-genetic TB risk factors. For example, while we tried to account for factors related to exposure by including only participants with a documented household exposure to an index TB patient, we do not have information on possible community or workplace exposures. We also could not correct for potential biases and unmeasured social discriminations and inequalities that might track with both genetic ancestry and TB progression risk. While these may be potential confounders, we consider it unlikely to explain the entirety of our signal: such biases are likely to track with households, and correction for household in our analyses did not alter our results. We note that the distribution of demographic variables such as sex, age, and education level in our cohort may differ from that in the general population. However, accounting for these covariates does not change our results, suggesting that our findings are unlikely to be driven by demographics. Finally, we emphasize that while our study brings proof that TB risk can vary across populations with different genetic ancestries, our results cannot be generalized to all indigenous populations, as different populations have different histories, and the sociodemographic factors, the primary determinants of TB risk, vary widely across different indigenous populations.[2,28]

## STAR★METHODS

Detailed methods are provided in the online version of this paper and include the following:

- KEY RESOURCES TABLE
- RESOURCE AVAILABILITY
  - Lead contact
  - Materials availability
  - Data and code availability
- EXPERIMENTAL MODEL AND SUBJECT DETAILS
  - Study participants
  - Phenotype description
- METHOD DETAILS
  - Categorizing smoking, drinking, body mass index (BMI), and socioeconomic status
  - Genotyping and global ancestry inference
  - Kinship estimation and genetic relatedness matrix (GRM)

**Table 3. Association of genetic ancestry and TB progression risk**

| Model | Genetic ancestry | $OR_{NAT0.1}$ (CI) | p value |
|---|---|---|---|
| NAT + age + sex + SES + (1\|HH) + GRM | NAT | 1.25 (1.18–1.33) | $1.1 \times 10^{-13}$ |
| EUR + age + sex + SES + (1\|HH) + GRM | EUR | 0.76 (0.70–0.83) | $1.3 \times 10^{-10}$ |
| AFR + age + sex + SES + (1\|HH) + GRM | AFR | 0.70 (0.60–0.80) | $8.5 \times 10^{-7}$ |
| ASI + age + sex + SES + (1\|HH) + GRM | ASI | 0.77 (0.59–1.00) | 0.05 |
| NAT + EUR + age + sex + SES + (1\|HH) + GRM | NAT | 1.24 (1.11–1.39) | $2.5 \times 10^{-4}$ |
| | EUR | 0.99 (0.84–1.16) | 0.89 |
| NAT + AFR + age + sex + SES + (1\|HH) + GRM | NAT | 1.24 (1.15–1.34) | $8.5 \times 10^{-8}$ |
| | AFR | 0.97 (0.81–1.16) | 0.73 |
| NAT + ASI + age + sex + SES + (1\|HH) + GRM | NAT | 1.26 (1.18–1.34) | $6.5 \times 10^{-13}$ |
| | ASI | 1.11 (0.85–1.45) | 0.44 |

Native Peruvian genetic ancestry (NAT) is associated with TB progression risk while European (EUR), West African (AFR), and East Asian (ASI) genetic ancestries were associated with reduced TB progression risk. NAT remained significantly associated with increased TB progression risk conditioned on non-NAT ancestries, but none of the non-NAT ancestries showed association with TB progression risk after conditioning on NAT. Odds ratios and 95% confidence interval (CI) correspond to 10% increase in genetic ancestry ($OR_{NAT0.1}$). p, two-sided Wald test p value; SES, socioeconomic status; HH, household.

○ Testing the association between global genetic ancestry proportions and TB progression risk
○ Testing the association between self-reported race and TB progression risk
○ Local ancestry inference
○ Admixture mapping
● QUANTIFICATION AND STATISTICAL ANALYSIS

**SUPPLEMENTAL INFORMATION**

**ACKNOWLEDGMENTS**

We thank all participants enrolled in this study. The study was supported by the National Institutes of Health (NIH) TB Research Unit Network, grants U19-AI111224-01 and U01-HG009088, and NIH grants U01-HG009379 and 1R01AR063759. The content is solely the responsibility of the authors and does not necessarily represent the official views of the NIH. S.A. was supported by the Swiss National Science Foundation postdoctoral mobility fellowships P2ELP3_172101 and P400PB_183823 and NIH T32 grant T32HG010464.

**AUTHOR CONTRIBUTIONS**

S.R., M.B.M., and S.A. designed the study. S.A. conducted the primary statistical analyses. Y.L., C.-C.H., and Z.Z. performed additional statistical analyses. D.J. interpreted data in the context of social and demographic features. M.B.M., L.L., R.C., J.J., R.Y., C.C., and J.T.G. recruited patients and obtained samples for this study. M.B.M. and D.B.M. established the collaborative framework in Boston for this multidisciplinary study. S.A. and S.R. drafted the initial manuscript. All authors discussed the results and commented on the manuscript.

**DECLARATION OF INTERESTS**

The authors have no conflict of interests.

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

## Article

**CellPress**

# STAR★METHODS

## KEY RESOURCES TABLE

| REAGENT or RESOURCE | SOURCE | IDENTIFIER |
| --- | --- | --- |
| **Deposited data** | | |
| Individual-level TB status and genotyping data | Luo et al., 2019 | phs002025.v1.p1 |
| **Software and algorithms** | | |
| Plink | Chang et al., 2015 | https://www.cog-genomics.org/plink/ |
| PCAdmix | Brisbin et al., 2012 | https://sites.google.com/site/pcadmix/downloads/copyright_1-0 |
| ADMIXTURE | Alexander et al., 2009 | https://dalexander.github.io/admixture/download.html |
| GENESIS R package | Conomos et al., 2016 | https://bioconductor.org/packages/release/bioc/html/GENESIS.html |
| lme4qtl | Ziyatdinov et al., 2018 | https://github.com/variani/lme4qtl |
| **Other** | | |
| The 1000 Genomes Project | Consortium, 2015 | https://www.internationalgenome.org/home |
| Genotyping data from Native Siberian and Native Americans | Reich et al., 2012 | Direct communication with authors |

## RESOURCE AVAILABILITY

### Lead contact
Further information and requests for resources and data should be directed to and will be fulfilled by the lead contact, Soumya Raychaudhuri (soumya@broadinstitute.org).

### Materials availability
This study did not generate new unique reagents.

### Data and code availability
Individual-level TB status and genotyping data are available through dbGAP (dbGAP: phs002025.v1.p1). No custom code was used to draw the central conclusions of this work. All the software and packages used in this work are included and referenced in the manuscript. Any additional information required to reanalyze the data reported in this paper is available from the lead contact upon request.

## EXPERIMENTAL MODEL AND SUBJECT DETAILS

### Study participants
We obtained blood samples for genotyping with informed consent from participants and following approval by the Institutional Review Board of Harvard School of Public Health (Reference number 19332) and by the Research Ethics Committee of the National Institute of Health of Peru.

The individuals included in the current study are a subset of a much larger cohort that we have collected and described in detail previously.[29,30] We recruited participants from 106 district health centers in Lima, Peru in a catchment area encompassing 12 of the 43 districts of metropolitan Lima and 3.3 million inhabitants (Figure S2). In this study, we included the subset of this larger study that were HIV-negative TB patients and HHCs and provided a blood sample. There are 3425 individuals in the final cohort including 43% females (N total = 2015 patients with active TB and 1320 HHCs with latent TB). The average age in our cohort is 31.29 years old (sd = 15.23).

### Phenotype description
We define cases as those individuals with confirmed TB diagnosis by the presence of acid-fast bacilli in a sputum smear, a positive *M. tb* culture, or based on diagnosis by a physician. We performed mycobacterial interspersed repetitive units (MIRU) based genotyping on all cultured *M. tb* isolates. Index cases are the first TB cases within each household.

We define controls as HHCs of index cases (i.e. individuals who live in the same house as the index case) who are infected with *M. tb* but did not progress to active diseases through the one-year follow-up. We determined the infection status of HHCs using TST and evaluated them for signs and symptoms of pulmonary and extra-pulmonary TB disease at baseline, and at two, six, and

 CellPress

12 months after enrollment. While cases were not excluded based on TB history, controls with a history of active TB or previous positive TST were excluded. We chose this study design because HHCs of individuals with TB are highly exposed to *M.tb* and are at a high risk of developing TB[31,32]; hence our strategy allowed us to focus on TB progression by including controls that were exposed to the pathogen and were infected. To adjust for any residual confounding that might be missed by our household recruitment study design, we also collected extensive sociodemographic and clinical variables at baseline including self-reported race and ethnicity, age, sex, body mass index (BMI), smoking, alcohol use, previous TB, socioeconomic status, education level, and BCG vaccination.

We refer to HHCs who receive a TB diagnosis within 14 days of enrollment of index cases as "baseline" cases. We refer to HHCs that developed active TB (e.g. became TB cases) 14 days or more after index case enrollment as "secondary cases" and to secondary cases that their *M. tb* strain shared exact MIRU genotyping with another TB case as "secondary clustered cases". In analyses focused on secondary or secondary clustered cases, controls were restricted to HHCs of these cases (i.e. if a household did not have any secondary or secondary clustered cases HHCs from that household were not included as controls).

## METHOD DETAILS

### Categorizing smoking, drinking, body mass index (BMI), and socioeconomic status

We categorized participants according to their alcohol intake as follows: *nondrinkers* if they reported having consumed no alcoholic drinks per day, *light drinkers* if they reported drinking <40 g or <3 alcoholic drinks per day and *heavy drinkers* if they reported drinking 40 g of alcohol or more or 3 or more drinks per day.[33] For smoking, we classified people as *nonsmokers* if they reported no cigarette smoking, as *light smokers* if they reported smoking one cigarette per day, and as *heavy smokers* if they reported smoking more than one cigarette per day.[33] We categorized people with BMI *z*-scores of less than −2 as *underweight* and those greater than 2 as *overweight*. For children, we defined the nutritional status based on the World Health Organization BMI z-score tables.[34] We calculated household-level composite socioeconomic scores (SES) using principal component analysis (PCA) as described before[30] by summarizing the following household-level factors: type of housing, the total number of rooms in the house, exterior wall material, primary floor material, primary roof material, type of water supply, type of sanitation facility, and type of lighting in the house. We categorized the continuous SES scores into tertiles corresponding to *low*, *middle*, and *upper* socioeconomic status groups.

### Genotyping and global ancestry inference

We extracted genomic DNA from participants' whole blood. To optimally capture the genetic diversity of Peruvians, we designed a customized array (LIMAArray) with 712,200 markers. In addition to the general genome-wide markers from the Affymetrix Axiom® my-Design custom genotyping array, we supplemented our array using coding markers from exome sequencing data of 116 Peruvian TB cases from the same population as our study population in order to optimally capture Peruvian's genetic variation, and particularly rare and protein-coding variations[7] (Method S1). Our array included 1.6% coding and 98.4% non-coding SNPs. Following QC and filtering, we kept 677,232 genotyped variants to use for downstream PCA and genetic ancestry inference. We merged genotyping data from our cohort, with previously published data from the 1000 Genomes Project phase 3 (2,054 individuals from 26 populations)[35] and Siberian and Native American populations from Reich et al.[36] (493 individuals from 57 native American populations and 245 individuals from 17 Siberian populations), by matching on the chromosome, position, reference, and alternate alleles using PLINK (version 1.90b3w).[37] After merging the datasets, we excluded variants with an overall minor allele frequency (MAF) < 1%. We then pruned the data for linkage disequilibrium (LD) by removing the markers with $r^2 > 0.1$ with any other marker within a sliding window of 50 markers per window and an offset of 10 using PLINK. The final merged dataset included 22198 variants (Methods S2, S3, and S4). We used the Genome-wide Complex Trait Analysis tool (GCTA),[38] version 1.26.0) to perform PCA and ADMIXTURE[39] (version 1.3), an unsupervised clustering method, with K = 4-7 clusters to perform global ancestry inference on this dataset (Figure S3). We used reference populations in ADMIXTURE analysis to determine what genetic ancestry each cluster represents. For example, if a cluster was the dominant cluster in the European individuals from the 1000 Genomes Project[35] we concluded that this cluster represents European genetic ancestry in admixed Peruvians from our cohort. All genetic analyses were done using GRCh37.

### Kinship estimation and genetic relatedness matrix (GRM)

We used the PC-Relate[40] implemented in the GENESIS R package (version 2.6.1) to estimate the kinship coefficients between individuals and to generate a genetic relatedness matrix (GRM). We removed rare variants (MAF <1%), regions with known long-range linkage disequilibrium (LD),[41] and variants in high LD ($r^2 > 0.2$ in a window of 50 kb and an offset of 5) using Plink (version 1.90b3w). In total, 551 pairs had kinship coefficients $\geq 0.125$ corresponding to second-degree relatives or closer. Of these, 36 pairs were parent-child, 72 were sib pairs, and 453 were second-degree relatives. Of related pairs, we randomly removed one individual. In total 430 individuals were removed. The remaining cohort included 1,929 TB cases and 1,066 HHCs. All remaining pairwise relatedness estimates were <0.125 (Figure S6).

### Testing the association between global genetic ancestry proportions and TB progression risk

To test the association of genetic ancestry proportions with TB progression risk we used the following logistic mixed model framework implemented in lme4qtl[42] R package:

$$\log\left[\frac{y_i}{1-y_i}\right] = \theta + \beta_{age}X_{d,age} + \beta_{sex}X_{d,sex} + \beta_{anc}X_{d,anc} + \beta_{ses}X_{d,ses} + (1|household) + GRM \quad \text{(Equation 1)}$$

$Yi$ is the probability of individual $i$ being a case, $\theta$ is the intercept, $\beta_{age}$ is a vector of effect estimates for individual-level standardized age {0–1}, $\beta_{sex}$ is a vector of effect estimates for individual-level sex {male, female}, $\beta_{nat}$ is a vector of effect estimates for individual-level native Peruvian, European, West African, or East Asian genetic ancestry proportions {0–1}, $X$s are the corresponding values for individual $d$, as appropriate. *household* is a categorical random effect for the household each individual comes from and GRM is a matrix of pairwise genetic relatedness values between individuals {0–0.5}.

We also performed the following sensitivity analyses to test the effect of additional covariates, phenotypic heterogeneity, or factors related to exposure or transmission on our results. First, in addition to native Peruvian genetic ancestry, age, sex, socioeconomic status, household, and GRM, we also included the following individual-level covariates in the above model: West African and East Asian genetic ancestries BMI, smoking, alcohol use, previous TB, education level, and BCG vaccination. Second, we restricted the analysis to 2043 microbiologically confirmed TB cases and 950 HHCs who were TST positive at baseline and did not develop active TB during the one year follow up and tested the association of native Peruvian genetic ancestry and TB progression risk using Equation 1. Third, we restricted the analysis to secondary cases (N = 213) and their HHCs (N = 214) and tested the association of native Peruvian genetic ancestry and TB progression risk using Equation 1. Finally, we performed a sensitivity analysis using 58 secondary clustered TB cases and their 48 HHCs and tested the association of native Peruvian genetic ancestry and TB progression risk using Equation 1.

For all above analyses, we removed individuals with any missing values for the included covariates. For all analyses, we calculated z-score for given covariates and used Wald test to calculate a two-sided p value.

## Testing the association between self-reported race and TB progression risk

To test the association between self-reported race and TB progression we used the following logistic regression model framework implemented in R's glm function:

$$\log\left[\frac{y_i}{1-y_i}\right] = \theta + \beta_{self-reported\ race}X_{d,\ self-reported\ race} \quad \text{(Equation 2)}$$

$Yi$ is the probability of individual i being a case, $\theta$ is the intercept, $\beta_{self-reported\ race}$ is a vector of effect estimates for self-reported race (categorical variable with eight categories).

## Local ancestry inference

"Local ancestry" is defined as the genetic ancestry of an individual at a particular locus, where an individual can have 0, 1, or 2 copies of an allele derived from each ancestral population.[43] We performed local ancestry inference using PCAdmix,[44] using imputed data to increase the number of shared markers between our data and reference data. Following phasing and imputation as described previously,[7] we excluded SNPs with imputation quality score $r^2 < 0.4$, HWE p value $< 10^{-5}$ in controls, or a missing rate per SNP greater than 5% which left 7,756,401 markers. To increase the number of overlapping variants between our cohort and the reference panel, we chose reference individuals with whole-genome sequencing data available including 25 native American individuals from the Simons Genome diversity project[45] plus 5 individuals from the 1000 Genomes Project[35] PEL that had inferred native Peruvian genetic ancestry >0.95 based on ADMIXTURE analysis at K = 4 clusters as proxy for native Peruvian genetic ancestry, 30 randomly selected individuals from the 1000 Genomes study CEU as proxy for European genetic ancestry, 30 randomly selected individuals from the 1000 Genomes study YRI as proxy for West African genetic ancestry, and 30 randomly selected individuals from CHS population as proxy for East Asian genetic ancestry. We then merged our data with the reference panel data and restricted the merged dataset to variants with MAF >5% in each of the reference populations. The post-QC dataset included 2,910,169 variants. We phased the merged data using SHAPEIT2.[46] After phasing and following PCAdmix developer's recommendation, we used PLINK (version 1.90b3w) to remove the markers with $r^2 > 0.8$ with any other marker within a sliding window of 20 markers per window and an offset of 10 using. 889,203 variants after pruning remained for use in local ancestry inference. We then used SHAPEIT2[46] (version v2.r837) to generate VCF files followed by Beagle (version 4.1) to generate input files for PCAdmix. All files were generated per chromosome. Finally, we performed local ancestry inference on each chromosome using PCAdmix (version 3) with the following options -bed and -ld 0 and recombination maps from the 1000 Genomes Project.[47] Local ancestry inference was done in windows of 20 SNPs and in total local ancestry was inferred for 44470 intervals. Genomic regions with long-range LD[41] including the major histocompatibility complex were excluded for admixture mapping. To avoid noise introduced by potential phenotypic heterogeneity, we repeated our admixture mapping analysis using cases with microbiologically confirmed TB (N = 2043) and HHCs who were TST positive at baseline and did not progress to active TB over the one year follow up (N = 950) (Figure S8). For SNPs within windows with the lowest association p value, we report variant-level summary statistics (Table S13).

## Admixture mapping

We used admixture mapping, a method to associate the inferred ancestry of a locus with a trait in an admixed population,[48] to search for genomic loci that can explain some of the observed association between native Peruvian genetic ancestry and TB progression.

**Cell Genomics**
Article

We used the Generalized Mixed Model Association Test (GMMAT, version 1.0.3),[49] a generalized linear mixed model framework to check the association between the inferred local native Peruvian genetic ancestry {coded as a number between 0 and 2 for local native ancestry posterior probability, 0 means no native Peruvian allele and 2 means 2 native Peruvian alleles} and TB progression risk {case, control}. We included standardized age {0–1}, sex {male, female}, global EUR, AFR, and ASI genetic ancestry proportions {0–1}, and a matrix of pairwise genetic relatedness {0–0.5} as covariates in the model. While the total number of loci tested was 44,470 we recognized that adjacent markers were highly correlated and that Bonferroni correction would be too stringent. Hence, to define the significance threshold for admixture mapping we permuted the case-control status and repeated the association analysis 1000 times. We then used the lowest p value from each permutation to generate an empirical null distribution. The fifth percentile of this distribution was used as the cutoff for genome-wide significance.

## QUANTIFICATION AND STATISTICAL ANALYSIS

All the statistical methods and softwares used in this study are listed in the corresponding sections in the method details. The statistical significance was determined by properly accounting for multiple testing as described in the method details. All p *value*s are two-sided.

