## [Document S2. Transparent peer review records for Asgari et al · Cell Genomics]

Higher native Peruvian genetic ancestry proportion is associated with tuberculosis progression risk

Author list

Samira Asgari¹⁻⁵, Yang Luo¹⁻⁵, Chuan-Chin Huang⁶⁻⁸, Zibiao Zhang⁶⁻⁸, Roger Calderon¹⁰, Judith Jimenez¹⁰, Rosa Yataco¹⁰, Carmen Contreras¹⁰, Jerome T. Galea^{6,11}, Leonid Lecca¹⁰, David Jones^{6,9}, D. Branch Moody², Megan B. Murray^{*6-8}, Soumya Raychaudhuri^{*1-5,12}

Summary

Initial submission: Received : May 22, 2021

Scientific editor: Orli Bahcall, Rosalind Mott

First round of review: Number of reviewers: 3
Revision invited : October 11, 2021
Revision received : Nov 29, 2021

Second round of review: Number of reviewers: 3
Accepted :

Data freely available:

Code freely available:

This transparent peer review record is not systematically proofread, type-set, or edited. Special characters, formatting, and equations may fail to render properly. Standard procedural text within the editor's letters has been deleted for the sake of brevity, but all official correspondence specific to the manuscript has been preserved.

Referees' reports, first round of review

Referee #1

The manuscript entitled "Higher native Peruvian genetic ancestry proportion is associated with tuberculosis progression risk" by Ashari et al reports an association study between TB and genetic ancestry using 2105 TB cases and 1320 household contacts. Samples were genotyped using a custom array based on whole exome sequencing data. After merging with reference populations, 22198 variants were available for analysis. Analyses were adjusted for population structure (including relatedness), sociodemographic factors and exposure or transmission influences. Indigenous Peruvian genetic ancestry was associated with TB progression risk independently of these factors and those with the highest proportion of indigenous Peruvian ancestry were three times more likely to progress to active disease. The results indicate a polygenic architecture for this association, as no single genome-wide locus could be identified. In addition, the results are specific to Peruvians and cannot be generalised across other indigenous population, who have their own histories, Mtb exposures and sociodemographic influences.

The data analyses were performed with technical rigor and well-explained, but I am unable to judge the genotyping quality control as this was not described in details. The figures and tables support the findings presented in the manuscript. I found the paper interesting, as it adds to the body of evidence that TB susceptibility is polygenic and population-specific.

Major Comments:

1. Please provide a reference for this statement - Line 105: "HHCs of individuals with TB are much more highly exposed to M.tb than those in the community; hence our strategy has the effect of controlling for environmental factors related to exposure." This statement is not strictly true - HHC studies lack a mechanism for accounting for community transmission, which is known to be significant, particularly in high burden settings (for example PMID: 31639145 and PMID:10325918)
2. Given that HHC were used, it is expected that there would be related individuals in the cohort and this was adjusted for in the analyses. What happens when only unrelated individuals are included in the analysis?
3. Why was only Peruvian ancestry modelled in the admixture section?
4. After merging with the reference population, the SNP density is quite low (22,198). Which QC steps and filtering was performed e.g Hardy-Weinberg equilibrium cut-offs etc.
5. What is the effect of using whole exome sequencing data for the design of the array? Many regulatory SNPs have been associated with TB susceptibility in other studies.
6. The authors state that African ancestry is protective in their cohort. I suggest to state that West African ancestry is protective instead - Africa is a large continent with significant genetic diversity.

Minor comments:

7. Although I am sure the appropriate IRB approvals were obtained, none of the approval numbers are listed in the manuscript.
8. Line 52: "mycobacterium" should be "Mycobacterium"
9. Line 172: "We also performed a the following sensitivity analyses". Please correct
10. Line 187: Incomplete sentence.
11. Line 423: Word missing. "HHCs that were TST positive for the but"

Referee #2

Asgari and colleagues reports on an ancestry specific association study with tuberculosis (TB) in an admixed Peruvian population. The authors quantified the global genetic ancestry of 2105 TB patients and 1320 household contacts. Association analysis between genetic ancestry and TB revealed a strongly significant association between native Peruvian ancestry and TB risk. Following admixture mapping didn't reveal significant associated loci.

Major concern:

Estimates of the different ancestries are strongly dependent on each other. As shown in table 1, European, African and Asian ancestry levels are strongly associated with the tertile of the native Peruvian genetic ancestry distribution. While it is clear that global ancestry is strongly associated with the risk of TB in this dataset, I'm wondering if we can really incriminate native Peruvian ancestry rather than European or African ancestries. How did the authors account for the correlation between estimates of ancestry? In Table 2,

strong p-value increase (or decrease of the significance level) for association of Native Peruvian ancestry (NAT) and European ancestry (EUR) with TB in the regression model including both variables (model 5: p-value NAT = 0.00025 and p-value EUR = 0.89) compared to the models including only one of the two (models 1: p-value NAT=10-13 and model 2: p-value EUR=10-10) suggest to me substantial collinearity/correlation between NAT and EUR. Not accounting for collinearity between the dependent variables in multivariable regression analysis can lead to misleading results. The most significant predictor (NAT) in "single ancestry" models may not be the causal one. If we take the example of SNP-based association studies, it is well known that the top associated SNP in a given locus has good chance not to be the causal one, even if the causal SNP was genotyped. Similar phenomenon cannot be ruled out here. In order to draw solid conclusions, I strongly advice the authors to use appropriate methods to deal with correlated variables.

Other comments:

In the admixture mapping analysis, global European, African and Asia ancestry proportions were included in the regression model. What is the rationale/hypothesis behind? Again, I'm wondering what is the impact of the correlation structure between those covariables +/- local ancestry on the association results.

Chimusa et al. (Hum Mol Genet, 2014) showed that local ancestry inference may be inaccurate in multi-way admixed populations. It should be acknowledged in the discussion.

It's not clear who were exactly the cases and controls used in this study. Cases include index TB case and secondary cases? Controls = all household contacts (HHC)? Only infected HHCs? How was defined infection

Referee #3

Historically, indigenous American populations have high rates of TB. The authors speculate that this high rate may be due to the presence of TB susceptibility alleles that have undergone negative selection in other populations that were historically more exposed to TB. To test this hypothesis, the authors report results of genetic admixture analysis in a Peruvian population of TB cases and controls. They find greater indigenous Peruvian genetic ancestry in the cases versus the controls, concluding that the high prevalence in indigenous Peruvians (albeit perhaps not in other indigenous groups) is genetically determined.

While interesting, there are concerns about the basic study design and analyses performed that should be addressed to make a more convincing case.

The authors state early on the following:

"First, our study uses genetic data to quantitatively assign genetic ancestry whereas previous studies used self-reported ancestry which is often a poor proxy for genetic ancestry and thus can lead to misclassification of participants."

I presume by self-reported ancestry the authors mean race and/or ethnicity. Such self-description is not a proxy for anything but what it is. Genetic ancestry estimates, at least in this case, the proportion of genetic material inherited from ancestors from different continents. The authors have not stated the race and/or ethnicity of the subjects in this cohort and how genetic ancestry might correlate with it. For example, do the individuals with greater than 90% indigenous ancestry self-report as indigenous? How do the individuals with less than 50% indigenous ancestry self-report? This is important because of potential confounding. In the results of Table 1, it appears that the indigenous ancestry difference between cases and controls only occurs in tertiles 2 and 3 of indigenous ancestry and not tertile 1. Tertiles 2 and 3 have high indigenous ancestry. How do those individuals self-report?

On a related topic, the authors note geographic correlates with genetic ancestry. Does that geography therefore also reflect differences in racial self-reporting?

Because TB is an infectious disease, it is important to characterize and control for infection in considering host characteristics. To this end, the authors have chosen a design to select controls from the same households that active TB cases derive from. As the authors state, choosing controls from households controls for possible confounders. That might be true, but depends completely on how the controls are chosen. This is not a matched design and not analyzed as such. It is clear that it is not one household control for each case, especially because there are many more cases than controls. So there is really no way to know or conclude that the controls are matched for confounders.

Furthermore, among the controls some active cases of TB are identified. These are considered secondary cases and included among the cases for most analyses, except for one that compares these secondary cases to baseline cases, where no difference in ancestry is discovered. The authors argue that the controls are matched to the cases on exposure. However, I can't see how that is the case, because it appears they are

suggesting that the "controls" were exposed at home while the cases were presumably exposed elsewhere. This also pertains to the analysis that compared a small number of baseline cases to secondary cases from the same households. We have no way to know that the exposure and length of exposure was comparable at all.

One way to gain some insight about household exposure is to compare the TB strain between the cases and their household controls. These data are not directly provided, but in one analysis that matched for strain, the numbers of cases and controls was very small, so one might wonder whether the others were not matched and hence had different exposures.

Little description is given of who the household controls are. It is typical that biological family members occupy the same household. The authors do comment on utilizing a genetic correlation matrix in the analysis, but they do not report the relationship status overall between the cases and controls. For example, how many parent-child pairs are there? How many sib pairs? How many spouse pairs? And who are the others that don't share one of these relationships? Of note, sibs are virtually identical for genetic ancestry; parents and children possibly not, but tend to be similar.

The authors end up with about 22K markers for their analysis. It appears these markers are derived from an array that was derived from exome sequencing of indigenous Peruvians. While this number may be adequate for global ancestry estimation, it is far too sparse for admixture mapping - so it is not clear what the real power was of the admixture mapping analysis that was done. In any event, there were no significant results from the admixture mapping. The authors however, chose to highlight a locus on chromosome 5 that was not significant. The authors have previously reported a genome wide association study of the same subjects, with no results in this region in the admixture mapping. There was a single locus identified in the GWAS on chromosome 3, but there is no signal here. There appears to be an allele frequency difference between indigenous Americans (low frequency) and Europeans (about 5-6%), but perhaps that's not a big enough difference for a signal (although the other paper reports that progression to TB in this population is driven by the locus on chromosome 3).

The authors have included a sizeable number of covariates in their analysis, which is a plus. However, they never report whether these covariates are associated with TB progression and their significance. Of possible concern, the authors report no correlation of their measure of SES (based on home features) with genetic ancestry, while education level is inversely correlated with indigenous ancestry. It is not clear why these are not tracking similarly.

For the admixture analysis, it appears to be an unsupervised analysis that includes a range of indigenous American populations, not just Peruvians. Yet the authors report results as Peruvian indigenous ancestry rather than Native American ancestry. It appears that many different indigenous groups were used in the estimation, so this may be a bit misleading.

At one point, the authors state the following:

"Second, we restricted the analysis to 2043 microbiologically confirmed TB cases and 950 HHCs who were TST positive at baseline and did not develop active TB during the one year follow up and tested the association of native Peruvian genetic ancestry and TB progression risk using equation 1. Third, we restricted the analysis to 2043 microbiologically confirmed TB cases and their 950 HHCs to test the association of native Peruvian genetic ancestry and TB progression risk using equation 1."

I don't see a difference between what is second and third above. Can you clarify?

Table 1:

Not clear why you have provided P-values for ancestry when the tertiles were defined in terms of ancestry.

Age is significant, but U-shaped. That would suggest you might use age-squared in the regression analysis also.

Females appear to have much more indigenous ancestry than the males. Why is that the case? And how does it relate to case-control status? Did you stratify your analyses by sex?

The authors report the following analysis:

"We considered that cases and controls who were from the same household may share a more similar ancestry profile compared to average which may bias our results. For this, we tested the association of native Peruvian genetic ancestry with TB progression risk using half of the cases (N=791) and the same number of controls who were not from the same household as cases and after correction for age, sex, socioeconomic status, and genetic relatedness. This analysis had a similar result to the analysis performed using the whole cohort (OR_{NATO.1}=1.19 (1.10-1.28), p-value=1.0x10⁻¹²)."

Including cases and controls that are matched on ancestry would have attenuated the OR. But here, it appears that the opposite is the case. Unclear why.

The authors state the following in their discussion:

"Our results suggest that clinicians and public health authorities in Peru should pay special attention to TB risk in areas where a large portion of the population are known to have higher levels of native Peruvian ancestry in order to ensure that the increased risk of TB progression in these individuals does not lead to increased health disparities."

What kind of special attention? Don't they already know those locations have high risk, irrespective of ancestry? Or are you suggesting that they test people's genetic ancestry in those locations and only attend to those with highest indigenous ancestry? Which health disparities are you talking about? Usually health disparities are defined in terms of race and/or ethnicity. Are you talking about the higher risk of TB among indigenous Peruvians? This really requires clarification.

In a related passage, the authors state:

"the importance of conducting large-scale host genetic studies in diverse populations in order to reduce health disparities and to get a comprehensive picture of the genotype-phenotype relationship in TB and other infectious diseases."

Again, the authors should clarify how the health disparity is defined, and precisely how this study will help reduce it.

Finally, I have a concern about novelty of this presentation. In the authors' previous GWAS paper published in Nature Communications, it appears that the same admixture figure, Figure 2a in the current manuscript, was previously published there in Figure 2a (or at least a pretty close facsimile thereof - one can easily see the admixture differences between cases and controls in the figure previously published, as is also reported here).

Comments enter in this field will be shared with the author; your identity will remain anonymous.

Authors' response to the first round of review

Reviewer #1: The manuscript entitled "Higher native Peruvian genetic ancestry proportion is associated with tuberculosis progression risk" by Asgari et al reports an association study between TB and genetic ancestry using 2105 TB cases and 1320 household contacts. Samples were genotyped using a custom array based on whole exome sequencing data. After merging with reference populations, 22198 variants were available for analysis. Analyses were adjusted for population structure (including relatedness), sociodemographic factors and exposure or transmission influences. Indigenous Peruvian genetic ancestry was associated with TB progression risk independently of these factors and those with the highest proportion of indigenous Peruvian ancestry were three times more likely to progress to active disease. The results indicate a polygenic architecture for this association, as no single genome-wide locus could be identified. In addition, the results are specific to Peruvians and cannot be generalised across other indigenous populations, who have their own histories, Mtb exposures and sociodemographic influences. The data analyses were performed with technical rigor and well-explained, but I am unable to judge the genotyping quality control as this was not described in details. The figures and tables support the findings presented in the manuscript. I found the paper interesting, as it adds to the body of evidence that TB susceptibility is polygenic and population-specific.

Major Comments:

1. Please provide a reference for this statement - Line 105: "HHCs of individuals with TB are much more highly exposed to M.tb than those in the community; hence our strategy has the effect of controlling for environmental factors related to exposure." This statement is not strictly true - HHC studies lack a mechanism for accounting for community transmission, which is known to be significant, particularly in high burden settings (for example PMID: 31639145 and PMID:10325918)

Answer: Reviewing the manuscript carefully, we now realize that a key aspect of our study design might have been under-emphasized. We would like to emphasize that in our study both our controls and cases are infected with M.tb; for controls who are the household contacts of cases, we confirmed infection status using TST status. Controls with a history of active TB or previous positive TST were excluded. Our study is focused on TB progression whereas many TB studies, including the ones mentioned by the reviewer, are focused on other aspects of TB such as transmission or infection dynamics. The fact that we only include HHCs that are TST+ allows us to ensure that our controls are not only exposed but also infected and thus lack of progression to active TB among HHCs is not due to lack of exposure or lack of infection. Hence transmission is not a confounding factor.

Our household contact study design allows us to better account for confounding factors other than exposure related to the living environment, for example, socioeconomic status and nutritional factors which are likely shared in the same household.

Previous studies show that close contacts of TB patients are at high risk for TB, especially within the first year (Martinez et al., 2017; Reichler et al., 2018). However, we agree with the reviewer that our study design does not account for community exposure and acknowledge this as a caveat in the discussion. We address this point by analyzing the ancestry difference between secondary cases who acquired TB through household contact and HHC controls (see Reviewer 3 comment #5)

To make the above points clearer we modified the sentence mentioned by the reviewer and also made the following changes to the manuscript text and added the above two references to the manuscript:

In the introduction:

We recruited TB patients and M.tb infected HHCs in whom we ascertained infection status by tuberculin skin testing (TST). We specifically picked controls in this way to make sure they are exposed and infected and to focus specifically on TB progression risk.

In methods:

We determined the infection status of HHCs using TST and evaluated them for signs and symptoms of pulmonary and extrapulmonary TB disease at baseline, and at six, and 12 months after enrollment (Figure 1). Controls with a history of active TB or previous positive TST were excluded. We chose this study design because HHCs of individuals with TB are highly exposed to M.tb and are at a high risk of developing TB (Martinez et al., 2017; Reichler et al., 2018); hence our strategy allows us to focus on TB progression by including controls that were exposed to the pathogen and were infected.

In discussion:

One caveat of our study is that we have not tested for all possible non-genetic TB risk factors. For example, while we tried to account for factors related to exposure by performing analysis on secondary cases, we do not have information on community and workplace exposures.

2. Given that HHC were used, it is expected that there would be related individuals in the cohort and this was adjusted for in the analyses. What happens when only unrelated individuals are included in the analysis?

Answer: We thank the reviewer for this suggestion. We have now revised the manuscript to include these results. In this new analysis, we used PC-relate (Conomos et al., 2016) to estimate the pairwise kinship coefficient between 3,425 individuals in our cohort. In total, 551 pairs had kinship coefficients ≥ 0.125 corresponding to second-degree relatives or closer. Of these pairs, we randomly removed one individual. In total 430 individuals were removed. The remaining cohort included 1929 TB cases and 1066 TST+ HHCs. Next, we repeated the analyses presented in Table 2 using this unrelated subset of individuals using the same covariates before and after removing the GRM. Removing related individuals did not change our results (Table S8 below).

Please also see the response to reviewer 3 comment #7 and accompanying new FigurebS6, where we use PC-relate results to investigate the type of relationships between related individuals in our cohort. (New) Table S8: Association of genetic ancestry and TB progression risk among unrelated individuals (N= 1943 TB cases and 1029 HHCs).

(New) Table S8: Association of genetic ancestry and TB progression risk among unrelated individuals (N= 1943 TB cases and 1029 HHCs).

Model	Genetic ancestry	OR _{NAT0.1} (CI)	P
NAT + age + sex + SES + (1 HH)	NAT	1.24 (1.17-1.31)	2.2x10 ⁻¹²
EUR + age + sex + SES + (1 HH)	EUR	0.78 (0.71-0.84)	2.2x10 ⁻⁹
AFR + age + sex + SES + (1 HH)	AFR	0.70 (0.60-0.81)	1.5x10 ⁻⁶
ASI + age + sex + SES + (1 HH)	ASI	0.77 (0.59-1.00)	0.05
NAT + EUR + age + sex + SES + (1 HH)	NAT	1.25 (1.11-1.40)	2.2x10 ⁻⁴
	EUR	1.01 (0.86-1.19)	0.91
NAT + AFR + age + sex + SES + (1 HH)	NAT	1.22 (1.13-1.32)	6.4x10 ⁻⁷
	AFR	0.95 (0.80-1.14)	0.59
NAT + ASI + age + sex + SES + (1 HH)	NAT	1.25 (1.17-1.33)	8.3x10 ⁻¹²
	ASI	1.09 (0.84-1.41)	0.53

We added the above table as Supplementary Table 8 and updated the manuscript text as follows to reflect this change:

...Next, to test whether these associations were independent of each other, we performed conditional analyses between ancestries. Native Peruvian genetic ancestry remained significantly associated with increased TB progression risk conditioned on the other ancestries, but the other ancestries showed no association with TB progression risk after conditioning on native Peruvian genetic ancestry (Table 2). Adjusting for self-reported race or removing 430 related individuals (kinship coefficient ≥ 0.125)

did not change these results (Figure S6, Table S7, Table S8).

We also updated the methods section for kinship estimation as follows:

We used the GENESIS R package (version 2.6.1) to estimate the kinship coefficients between individuals and to generate a genetic relatedness matrix (GRM). We removed rare variants (MAF < 1%), regions with known long-range linkage disequilibrium (LD) (Price et al., 2008), and variants in high LD ($r_2 > 0.2$ in a window of 50kb and an offset of 5) using Plink (version 1.90b3w). In total, 551 pairs had kinship coefficients ≥ 0.125 corresponding to second-degree relatives or closer. Of these, 36 pairs were parent-child, 72 were sib pairs, and 453 were second-degree relatives. Of related pairs, we randomly removed one individual. In total 430 individuals were removed. The remaining cohort included 1929 TB cases and 1066 HHCs. All remaining pairwise relatedness estimates were < 0.125 . The remaining cohort included 1,929 TB cases and 1,066 HHCs. All remaining pairwise relatedness estimates were < 0.125 .

3. Why was only Peruvian ancestry modelled in the admixture section?

Answer: We chose to model local native Peruvian genetic ancestry because it is the only ancestry that remained significantly associated with TB progression risk after accounting for other ancestries. However, to answer the reviewer's comment, we repeated our admixture mapping analysis by modeling local European ancestry coded as a number between 0 and 2 for local European ancestry posterior probability, 0 means no native Peruvian allele and 2 means 2 European alleles} and after correction for standardized age {0-1}, sex {male, female}, global native Peruvian, West African, and East Asian genetic ancestry proportions {0-1}, and a genetic relatedness matrix.

Given that native Peruvian and European ancestries are the main ancestries in our cohort (average proportions of 0.80 and 0.16 respectively) we expect that modeling local European ancestry instead of native Peruvian ancestry will give very similar results. As expected (Response letter Figure 1 below) the results are highly concordant with our previous results and the effect sizes for local European ancestry are inversely correlated with the effect sizes obtained from modeling local native Peruvian ancestry.

We did not model West African and East Asian ancestries; given their low global ancestry proportions, we expected the length of most tracks for these ancestries to be short and thus difficult to infer reliably (Gravel, 2012).

Response letter Figure 1: Admixture mapping modelling local European ancestry.

A) Admixture mapping using all cases and all HHCs (N = 2105 cases with active TB and 1320 HHCs with latent TB). We tested the association of local European ancestry with TB progression risk after correction for age, sex, global native Peruvian, West African, and East Asian ancestry proportions, and genetic relatedness (Right). Each dot represents a local ancestry interval including 20 SNPs (N = 44470 intervals). No loci passed the genome-wide significance threshold. Redline: 4.3×10^{-8} similar to the genome-wide significance threshold used for admixture mapping modeling local native Peruvian ancestry. The lowest p-value was at 5p23.2 (OR = 0.74 (0.64-0.85), p-value = 2.7×10^{-8}). We observed no inflation in the association p-values (left, quantile-quantile plot of admixture mapping p-values on the left, $\lambda = 0.93$). B) Admixture mapping Z scores were high concordant between the analysis modeling local native Peruvian and the analysis European ancestry; Pearson correlation coefficient ($r = 0.86$).

4. After merging with the reference population, the SNP density is quite low (22,198). Which QC steps and filtering was performed e.g Hardy-Weinberg equilibrium cut-offs etc.

Answer: We thank the reviewer for this comment. The total number of SNPs used for global ancestry inference is not large (22,198 post-QC SNPs) since it is the merger of multiple data sets: 1) our data, 2) Native American and native Siberians, and 3) 1000 Genomes Project. However, this SNP density is enough for reliably inferring global ancestry proportions. To highlight this point, we repeated our global ancestry analysis using post-QC data from our cohort plus the 1000 Genomes project data only (120,630 post-QC SNPs) and show that ancestry estimates are highly correlated for all individuals in our cohort between the two analyses (Figure S4 and Table S2 below). These results are in line with previous studies about the effect of the number of markers in the accuracy of global ancestry inference in admixed populations (Galanter et al., 2012; Pereira et al., 2020; Santos et al., 2016) which show that reliable global ancestry proportions can be obtained using as low as 120-200 ancestry informative markers.

Below we describe the QC steps used to generate the QCed dataset for PCA and global ancestry inference analyses.

Merging genotyping data from our Peruvian cohort with data from native American and Siberian individuals from Reich et al:

- 1- We used convertf to convert the original files including 364,470 variants to plink format.
- 2- We used liftOverPlink (<https://github.com/sritchie73/liftOverPlink>) to lift data from hg18 to GRCh37; 364,396 variants were lifted successfully.
- 3- We used plink v1.90b to update variant IDs to chromosome:position:allele1:allele2
- 4- We started from 677,232 bi-allelic variants and used plink v1.90b to update variant IDs to chromosome:position:allele1:allele2
- 5- We used plink --bmerge function to merge the Reich data and TBRU data
- 6- 9,907 variants failed merge due to having multiple alleles
- 7- We removed the failed variants and redid the merging
- 8- The merged file includes 994,444 variants
- 9- We applied the following QC measures on the merged file using plink v1.90b genotyping missingness < 5% (--geno 0.05) minor allele frequency > 1% (--maf 0.01) and Hardy-Weinberg equilibrium p-value > 10⁻⁵ (--hwe 10e-5) in controls applying these filters led to 34,958 post-QC variants that were present in both datasets
- 10- We then pruned the data for linkage disequilibrium (LD) by removing the markers with $r_2 > 0.1$ with any other marker within a sliding window of 50 markers per window and an offset of 10 using PLINK. The final merged dataset included 23,169 variants.

Merging genotyping data from our Peruvian cohort with data from native American and Siberian individuals from Reich et al and the 1000 Genomes Project:

- 1- We started from 81,083,551 bi-allelic variants from the 1000 Genomes Project phase 3 and used plink v1.90b to update variant IDs to chromosome:position:allele1:allele2
- 2- We then merged the 1000 Genomes Project variants with the 34,958 post-QC from the above analysis resulted in 81,117,189 variants
- 3- We applied the following QC measures on the merged file using plink v1.90b genotyping missingness < 5% (--geno 0.05) minor allele frequency > 1% (--maf 0.01) and Hardy-Weinberg equilibrium p-value > 10⁻⁵ (--hwe 10e-5) in controls applying these filters led to 34,936 post-QC variants that were present in both datasets
- 4- We then pruned the data for linkage disequilibrium (LD) by removing the markers with $r_2 > 0.1$ with any other marker within a sliding window of 50 markers per window and an offset of 10 using PLINK. The final merged dataset included 22,198 variants.

Merging genotyping data from our Peruvian cohort with data from the 1000 Genomes Project:

- 1- We started from 677,232 bi-allelic variants from our Peruvian cohort and 81,083,551 bi-allelic variants from the 1000 Genomes Project phase 3 with variant IDs in both files set to chromosome:position:allele1:allele2
- 2- We selected 521,779 the variants that were present in both datasets from each dataset
- 3- we merged the two datasets using the shared variants
- 4- We applied the following QC measures on the merged file using plink v1.90b genotyping missingness < 5% (--geno 0.05) minor allele frequency > 1% (--maf 0.01) and Hardy-Weinberg equilibrium p-value > 10⁻⁵ (--hwe 10e-5) in controls applying these filters led to 272,531 post-QC variants that were present in both datasets
- 5- We then pruned the data for linkage disequilibrium (LD) by removing the markers with $r_2 > 0.1$ with any other marker within a sliding window of 50 markers per window and an offset of 10 using PLINK. The final merged dataset included 120,630 variants.

(New) Figure S4: Global ancestry inference results using different reference panels. Correlation between global ancestry estimates using ADMIXTURE(Alexander et al., 2009) (K = 4) analysis with data from Peruvians included in this study (LIMAA, N = 3425) merged with the data from populations from the 1000 Genomes Project phase 3 (1KG, N = 2054)(Consortium et al., 2015) or the data from populations from the 1000 Genomes Project phase 3 plus Siberian and Native American populations(Reich et al., 2012) (Reich, N = 738) as the reference panel. Each dot represents one individual. r= Pearson correlation coefficient. NAT: native Peruvian ancestry, EUR: European ancestry, AFR: West African ancestry, ASI: East Asian ancestry.

Table S1: Global ancestry inference results using Global ancestry inference results using different reference panels.

input	# variants for analysis	NAT	EUR	AFR	ASI
LIMAA+1KG	120630	0.81(0.15)	0.15(0.11)	0.03(0.07)	0.01(0.03)
LIMAA+1KG+Reich	22198	0.80(0.15)	0.16(0.11)	0.03(0.07)	0.01(0.03)

ADMIXTURE(Alexander et al., 2009) (K = 4) analysis was done on data from Peruvians included in the study (LIMAA, N = 3425) merged with the data from populations from the 1000 Genomes Project phase 3 (1KG, N = 2054) (Consortium et al., 2015) and/or the data from Siberian and Native American populations (Reich et al., 2012) (N = 738) as the reference panel. The values are presented in the following format: average (standard deviation). NAT: native Peruvian ancestry, EUR: European ancestry, AFR: West African ancestry, ASI: East Asian ancestry.

We added a new supplementary figure (Figure S4) to the manuscripts to show the correlation of global ancestry estimates using different sets of markers and updated the manuscript's text to reflect this change as follows:

These four populations corresponded to native Peruvian, European, West African, and East Asian genetic ancestry with average proportions 0.80 (standard deviation (sd)=0.15), 0.16 (0.11), 0.03 (0.07), and 0.01 (0.03) respectively (Table 1, Figure 2A, Table S1, Figure S4).

We also added the above details about dataset merging to supplementary text and updated the methods section to reflect this change as follows:

After merging the datasets, we excluded variants with an overall minor allele frequency (MAF) $< 1\%$. We then pruned the data for linkage disequilibrium (LD) by removing the markers with $r_2 > 0.1$ with any other marker within a sliding window of 50 markers per window and an offset of 10 using PLINK. The final merged dataset included 22198 variants (Supplementary methods).

5. What is the effect of using whole exome sequencing data for the design of the array? Many regulatory SNPs have been associated with TB susceptibility in other studies.

Answer: We thank the reviewer for identifying this confusing point. We used exome sequencing data from 116 Peruvian TB cases from the same population as our study population in the design of our genotyping array (LIMAArray) in order to optimally capture Peru's genetic variation, and particularly rare and protein-coding variations. Our array consists of genome-wide content, including intronic and intergenic markers. When we designed this array, we combined the content from the Affymetrix Axiom. myDesign custom genotyping array with our exome sequencing data, and whole-genome sequencing data from 75 Peruvians from the 1000 Genomes project. Our array included 712,200 markers in total. Of these, only 1.6% are in the coding region, while the rest in the non-coding genome.

More details about the design of our array and its performance is presented in detail in our previous publication (Luo et al., 2019). We have now modified the manuscript text and added the above details about our genotyping array design to supplementary methods as follows:

To optimally capture the genetic diversity of Peruvians, we designed a customized array (LIMAArray) with 712,200 markers. In addition to the general genome-wide markers from the Affymetrix Axiom® myDesign custom genotyping array, we supplemented our array using coding markers from exome sequencing data of 116 Peruvian TB cases from the same population as our study population in order to optimally capture Peruvian's genetic variation, and particularly rare and protein-coding variations (Luo et al., 2019) (Supplementary methods). Our array included 1.6% coding and 98.4% non-coding SNPs.

We also added the following information to supplementary notes about array design:

Genotyping array design

We used exome sequencing data from 116 Peruvian TB cases from the same population as our study population in the design of our genotyping array (LIMAArray) in order to optimally capture Peru's genetic variation, and particularly rare and protein coding variations. We combined the content from the Affymetrix Axiom® myDesign custom genotyping array with our exome sequencing data, and whole-genome sequencing data from 75 Peruvians from the 1000 Genomes project. We then selected 302,400 markers including all coding variants, variants that were more frequent in Peruvians compared to other populations in the 1000 Genomes project, and variants in the proximity of known TB genes. Additionally, we included 8,100 markers from GWAS catalogue based on known or suggestive associations with autoimmune diseases, TB phenotypic variation, as well as markers from the HLA region. We also included 4,500 known ancestry informative markers. Finally, we included 397,200 markers that provide good genome-wide coverage for imputation in Peruvian populations in the common ($>5\%$), low frequency (1–5%) and rare (0.5–1%) MAF ranges, including many regulatory variants. All together this resulted in an array with 712,200 markers. For more details about LIMAArray and its performance see reference(Luo et al., 2019).

6. The authors state that African ancestry is protective in their cohort. I suggest to state that West African ancestry is protective instead - Africa is a large continent with significant genetic diversity.

Answer: We thank the reviewer for this suggestion and changed the “African ancestry” to “West African ancestry” throughout the manuscript. Similarly, we also changed “Asian ancestry” to “East Asian ancestry” throughout the manuscript to be more precise.

Minor comments:

7. Although I am sure the appropriate IRB approvals were obtained, none of the approval numbers are listed in the manuscript.

Answer: We added the IRB approval number and modified the text as follows:

The study protocol was approved by the Harvard University Institutional Review Board (Reference Number 19332) and by the Research Ethics Committee of the National Institute of Health of Peru.

8. Line 52: "mycobacterium" should be "Mycobacterium"

Answer: We corrected this mistake.

9. Line 172: "We also performed a the following sensitivity analyses". Please correct

Answer: We corrected this mistake.

10. Line 187: Incomplete sentence.

Answer: We corrected this mistake.

11. Line 423: Word missing. "HHCs that were TST positive for the but"

Answer: We corrected this mistake.

Reviewer #2: Asgari and colleagues reports on an ancestry specific association study with tuberculosis (TB) in an admixed Peruvian population. The authors quantified the global genetic ancestry of 2105 TB patients and 1320 household contacts. Association analysis between genetic ancestry and TB revealed a strongly significant association between native Peruvian ancestry and TB risk. Following admixture mapping didn't reveal significant associated loci.

Major concern:

1- Estimates of the different ancestries are strongly dependent on each other. As shown in table 1, European, African and Asian ancestry levels are strongly associated with the tertile of the native Peruvian genetic ancestry distribution. While it is clear that global ancestry is strongly associated with the risk of TB in this dataset, I'm wondering if we can really incriminate native Peruvian ancestry rather than European or African ancestries.

Answer: The reviewer raises an important point about the correlation between different ancestry proportions. We acknowledge that ancestry estimates in our cohort are correlated (Response letter Figure 2 below). Due to this correlation, we cannot be sure that it is native Peruvian genetic ancestry that increases TB progression risk versus European or African ancestry that decreases TB progression risk.

We chose to highlight native Peruvian ancestry for the following reasons: 1) In our cohort, native Peruvian is the main genetic ancestry and the only one that is associated with an increased TB progression risk relative to other ancestry components, 2) in univariate analysis it has the strongest level of association.

However, we also note that there is indeed an argument that the effects we see are because European ancestry is protective; recent genetic studies suggest that the high burden of TB among Europeans over the past 2000 years led to negative selection in this population and reduced the frequency of risk-increasing variance in Europeans (Kerner et al., 2021).

Despite this evidence, as the reviewer suggests, we cannot be sure of the causality of native Peruvian ancestry for TB risk vs. protectivity of European ancestry. To address this issue, throughout the paper we emphasize that native Peruvian ancestry is associated with increased TB progression risk “relative to other ancestries that we tested here” or “in comparison to native Peruvian genetic ancestry”.

Response letter Figure 2: Correlation between native Peruvian ancestry and other ancestries in our cohort (r : Pearson correlation coefficient). NAT: native Peruvian genetic ancestry, EUR: European genetic ancestry, AFR: West African genetic ancestry, ASI: East Asian genetic ancestry.

To further emphasize this issue in the discussion we updated the text as follows:

In our cohort, native Peruvian genetic ancestry is the only ancestry that remains significantly associated with TB progression risk after conditioning on other ancestries. In comparison to native Peruvian genetic ancestry, European and West African genetic ancestries were associated with reduced TB progression risk. This protective effect can be the result of the long, shared history of these populations with *M. tb* (Comas et al., 2013) which could have led to selective pressures that have mitigated TB genetic risk if such pressures were not present in pre-colonization Peru (Woodman et al., 2019). However, we want to emphasize that the effect of native Peruvian ancestry on TB progression risk is relative to other ancestries that we tested here and cannot be causally detangled from the effect of other genetic ancestries.

We did not add the above figure to the manuscript because, as noted by the reviewer, the correlation between ancestries is expected.

2- How did the authors account for the correlation between estimates of ancestry? In Table 2, strong p-value increase (or decrease of the significance level) for association of Native Peruvian ancestry (NAT) and European ancestry (EUR) with TB in the regression model including both variables (model 5: p-value NAT = 0.00025 and p-value EUR = 0.89) compared to the models including only one of the two (models 1: p-value NAT=10-13 and model 2: p-value EUR=10-10) suggest to me substantial collinearity/correlation between NAT and EUR. Not accounting for collinearity between the dependent variables in multivariable regression analysis can lead to misleading results.

Answer: We thank the reviewer for this comment. As mentioned above (reviewer 2 comment #1), our ancestry estimates are correlated ($r = -0.86, -0.67, -0.38$, Pearson correlation coefficient between native Peruvian and European, West African, and East Asian genetic ancestries respectively). However, we note that correlation between two covariates does not necessarily lead to multicollinearity and model instability (source: McElreath, R. (2020). *Statistical rethinking: A Bayesian course with examples in R and Stan*. 2nd edition. Chapman and Hall/CRC). Multicollinearity is usually diagnosed in two ways: a) by observing drastic changes in the estimated regression coefficients of covariates when adding or deleting a predictor, and b) by variance inflation factor analysis.

We performed both above analyses to determine if having both native Peruvian and European ancestry proportions in our regression model can cause problems. As the reviewer noticed, addition of European ancestry to the following model ($TB_status \sim NAT + age + sex + SES + (1|HH) + GRM$) leads to a lower p-value for NAT (1.1×10^{-13} vs. 2.5×10^{-4}) but the odds ratio for native Peruvian ancestry remains stable (1.25 (1.18-1.33) vs. 1.24 (1.11-1.29)).

We also calculated variance inflation factor for the following models ($TB_status \sim NAT + EUR + age + sex + SES + (1|HH) + GRM$ or $TB_status \sim NAT + AFR + age + sex + SES + (1|HH) + GRM$) as shown below in the Response letter Table 1 below, the variance inflation factor for all covariates in both models are < 5 indicating tolerable correlations between covariates. (source: James, G., Witten, D., Hastie, T., & Tibshirani, R. (2013). *An introduction to statistical learning: with applications in R*. New York: Springer., Page 102: “As a rule of thumb, a VIF value that exceeds 5 or 10 indicates a problematic amount of collinearity”).

Response letter Table 1: Variance Inflation Factor (VIF) analysis for the model including both native American and European ancestries.

Model terms	VIF	Model terms	VIF
Age	1.02	Age	1.02
Sex	1.02	Sex	1.02
SES	1.03	SES	1.03
NAT	3.94	NAT	1.80
EUR	3.93	AFR	1.79

We did not add the above table to the manuscript because the analyses results show that collinearity is not an issue in our statistical models; since we found the results largely reassuring, and recognize that this information would be shared through this response letter, which will be published alongside our manuscript.

3- The most significant predictor (NAT) in "single ancestry" models may not be the causal one. If we take the example of SNP-based association studies, it is well known that the top associated SNP in a given locus has good chance not to be the causal one, even if the causal SNP was genotyped. Similar

phenomenon cannot be ruled out here. In order to draw solid conclusions, I strongly advise the authors to use appropriate methods to deal with correlated variables.

Answer: Please see answers to reviewer 2 comments #1 and #2 above where we discuss the issue of correlation between genetic ancestry and its effect on our regression model. In summary, we agree that it is difficult to detangle the possibility that European genetic ancestry being protective against TB progression versus native Peruvian genetic ancestry being a risk factor.

Following the reviewer's comment, we make note of this point in the discussion:

However, we want to emphasize that the effect of native Peruvian ancestry on TB progression risk is relative to other ancestries that we tested here and cannot be causally detangled from the effect of other genetic ancestries.

Other comments:

4- In the admixture mapping analysis, global European, African and Asia ancestry proportions were included in the regression model. What is the rationale/hypothesis behind? Again, I'm wondering what is the impact of the correlation structure between those covariables +/- local ancestry on the association results.

Answer: By correcting for global ancestry proportions, we correct for the overall population structure (synonymous to correcting for PCs when performing SNP associations) in order to pick up local deviations in ancestry proportion between cases and controls. To demonstrate this point we repeated our admixture mapping without analysis without correcting for population structure, as shown below the p-values are inflated compared to the model that accounts for population structure by correcting for global ancestry proportions (Response letter Figure 3 below).

Response letter Figure 3: Admixture mapping QQ-plots with (left) and without (right) correction for global ancestry proportions.

Moreover, similar to the answer to reviewer 2 comment #1, we performed variance inflation analysis to ensure that our correlation between the covariates. Variance inflation factor for all covariates included in our admixture mapping model was < 5 (Response letter Table 2 below).

Response letter Table 2: Variance Inflation Factor (VIF) analysis for covariates included in our admixture mapping model.

Model terms	VIF
Age	1.02
Sex	1.02
SES	1.03
EUR	1.1
AFR	1.1

We did not add the above table to the manuscript because the analyses results show that inclusion of global ancestry as covariates in our admixture mapping is necessary and that collinearity is not an issue in our statistical models.

5- Chimusa et al. (Hum Mol Genet, 2014) showed that local ancestry inference may be inaccurate in multi-way admixed populations. It should be acknowledged in the discussion.

Answer: We added this reference to our discussion and modified the discussion as follows:

While our results show a strong genome-wide signal for the effect of native Peruvian genetic ancestry on TB risk, we did not identify any single locus that can explain this effect, suggesting that it is driven by a polygenic architecture with many variants exerting modest impact. Our power to detect specific risk loci through admixture mapping may have also been affected by the lower accuracy of local ancestry inference in multi-way admixture scenarios compared to two-way admixtures (Chimusa et al., 2014). Conclusively identifying such loci requires larger studies with greater statistical power.

6- It's not clear who were exactly the cases and controls used in this study. Cases include index TB case and secondary cases? Controls = all household contacts (HHC)? Only infected HHCs? How was defined infection

Answer: We thank the reviewer for bringing this point to light, and we have now recognized that the description around controls may have been confusing (see reviewer 1 comment #1).

To clarify, TB cases are the index TB cases (i.e. the first TB case in each household), baseline TB cases (i.e. individuals with TB who were identified within 14 days of index case enrollment), and secondary TB cases (i.e. cases who developed TB 14 days or more after index case enrollment). All TB cases have confirmed the TB diagnosis by the presence of acid-fast bacilli in a sputum smear, a positive M. tb culture, or based on diagnosis by a physician.

We define controls as HHCs of index cases (i.e. individuals who live in the same house as the index case) who are infected with M. tb but did not progress to active diseases through the one-year follow-up. Hence all controls were exposed and infected and shared a living environment with cases. We determined the infection status of HHCs using TST which is commonly used for establishing latent TB and evaluated

them for signs and symptoms of pulmonary and extrapulmonary TB disease at baseline, and at six, and 12 months after enrollment.

To make our case and control definitions clearer we updated the manuscript as follows

In the introduction:

We recruited TB patients and M.tb infected HHCs in whom we ascertained infection status by tuberculin skin testing (TST). We specifically picked controls in this way to make sure they are exposed and infected and to focus specifically on TB progression risk.

In methods under “Phenotype description”:

We define cases as those individuals with confirmed the TB diagnosis by the presence of acid-fast bacilli in a sputum smear, a positive M. tb culture, or based on diagnosis by a physician. We performed mycobacterial interspersed repetitive units (MIRU) based genotyping on all cultured M. tb isolates. Index cases are the first TB cases within each household.

We define controls as HHCs of index cases (i.e. individuals who live in the same house as the index case) who are infected with M. tb but did not progress to active diseases through the one year followup. We determined the infection status of HHCs using TST and evaluated them for signs and symptoms of pulmonary and extrapulmonary TB disease at baseline, and at six, and 12 months after enrollment (Figure 1). Controls with a history of active TB or previous positive TST were excluded. We chose this study design because HHCs of individuals with TB are highly exposed to M.tb and are at a high risk of developing TB (Martinez et al., 2017; Reichler et al., 2018); hence our strategy has the effect of controlling for low exposure rate, allowing allowed us to focus on TB progression by ensuring that all controls that were exposed to the pathogen and were infected. To adjust for any residual confounding that might be missed by our household recruitment study design, we also collected the extensive sociodemographic and clinical variables at baseline including age, sex, body mass index (BMI), smoking, alcohol use, previous TB, socioeconomic status, education level, and BCG vaccination (Table 1).

We refer to HHCs who receive a TB diagnosis within 14 days of enrollment of index cases as “baseline” cases. We refer to HHCs that developed active TB (e.g. became TB cases) 14 days or more after index case enrollment as “secondary cases” and to secondary cases that their M. tb strain shared exact MIRU genotyping with another TB case as “secondary clustered cases”.

Reviewer #3: Historically, indigenous American populations have high rates of TB. The authors speculate that this high rate may be due to the presence of TB susceptibility alleles that have undergone negative selection in other populations that were historically more exposed to TB. To test this hypothesis, the authors report results of genetic admixture analysis in a Peruvian population of TB cases and controls. They find greater indigenous Peruvian genetic ancestry in the cases versus the controls, concluding that the high prevalence in indigenous Peruvians (albeit perhaps not in other indigenous groups) is genetically determined.

While interesting, there are concerns about the basic study design and analyses performed that should be addressed to make a more convincing case.

The authors state early on the following:

"First, our study uses genetic data to quantitatively assign genetic ancestry whereas previous studies used self-reported ancestry which is often a poor proxy for genetic ancestry and thus can lead to misclassification of participants."

1- I presume by self-reported ancestry the authors mean race and/or ethnicity. Such self-description is not a proxy for anything but what it is. Genetic ancestry estimates, at least in this case, the proportion of genetic material inherited from ancestors from different continents. The authors have not stated the race and/or ethnicity of the subjects in this cohort and how genetic ancestry might correlate with it. For example, do the individuals with greater than 90% indigenous ancestry self-report as indigenous? How do the individuals with less than 50% indigenous ancestry self-report? This is important because of potential confounding.

Answer: We thank the reviewer for this important comment. In our cohort, we have access to self-reported race and ethnicity. The majority of individuals in our cohort self-identify their race as “American Indian + White” and self-identify their ethnicity as “Latino” (74% and 99% respectively, Table 2 below).

Following the reviewer's suggestion, we investigated the correlation between self-reported race and genetic ancestry in our cohort. Native Peruvian ancestry was the dominant genetic ancestry in all categories of self-reported race (Figure 2 below). We found that there are differences in genetic ancestry between individuals self-reporting different races.

Among individuals with > 0.9 native Peruvian ancestry ($N = 985$) 77% self-identified as “American Indian + White”, 18% as “American Indian”, and 4% as “White”; for self-reported ethnicity, 99% self-identified as Latino (Table S3 below). Among individuals with < 0.5 native Peruvian ancestry ($N = 140$) 57% self-identified as “American Indian + White”, 21% as “White”, 11% as “Black”, and 7% as “American Indian”; for self-reported ethnicity, 99% self-identified as Latino (Table S4 below). So in both groups with high (> 0.9) or low (< 0.5) proportion of native Peruvian genetic ancestry, the majority of individuals self-report their race as “American Indian + White” (77% vs. 57% respectively). However, a larger percentage of the group with a low proportion of native Peruvian genetic ancestry self-identify as “American Indian” (18% vs. 7%).

These results suggest that individuals who self-report the same can have drastically different levels of genetic ancestry proportions. Our results are in line with previous reports that self-reported race/ethnicity is a poor proxy for genetic ancestry in admixed populations (Kumar et al., 2010; Mersha and Abebe, 2015; Sucheston et al., 2012).

To assess if self-reported race predicted TB outcome we used the following model

TB status \sim self-reported race

No individual category of self-reported race was significantly associated with TB status (Table S6 below) and the above model accounting for multiple self-reported races in a multiple degree of freedom model was not significantly different from the following null model (likelihood ratio test $p = 0.73$):

TB status ~ 1

Altogether, these results suggest that in our cohort self-reported race is not a risk factor for TB progression.

Finally, to test if the racial group that individuals self-identify with affect our association results we repeated our analysis using the following model:

TB status \sim age + sex + NAT + self-reported race + (1|HH) + GRM

Where NAT is native Peruvian genetic ancestry proportion, HH is household, and GRM is a genetic relatedness matrix. Native Peruvian ancestry remained significantly associated with increased TB progression risk (1.26 (1.18-1.33), $p=2.3 \times 10^{-13}$; Table S7 below). This model is not significantly different from the model that does not include race (F-test, $p=0.87$), confirming little to no association with TB progression is explained by self-reported ancestry.

Altogether, these results show that self-reported race and genetic ancestry are correlated; however, individuals who self-report the same can have drastically different levels of genetic ancestry proportions. Additionally, our results suggest that in our cohort, self-reported race is not a risk factor for TB progression and cannot explain the association between genetic ancestry and TB progression risk.

(New) Table 2: Self-reported race and ethnicity in our cohort (N = 3,425)

Self-reported race	Count
American Indian + white	2538
American Indian	515
White	289
Black	42
American Indian + Black	15
Asian	6
Black + White	6
American Indian + White + Asian	1
NA	13
Self-reported ethnicity	Count
Latino	3410
Not Latino	9
NA	6

(New) Figure 2: Genetic ancestry proportions among different self-reported race categories. Native Peruvian genetic ancestry (NAT) is the dominant genetic ancestry in all categories of self-reported race. The proportion of NAT, European (EUR), West African (AFR), and East Asian genetic (ASI) ancestries was significantly different between self-reported race categories (ANOVA, $df=7$, p -value = 1.4×10^{-47} , p -value = 1.7×10^{-48} , 1.6×10^{-32} , 1.2×10^{-118} , and 8.2×10^{-32} respectively respectively). Box plots show median and interquartile range (IQR) and whiskers show $1.5 \times IQR$.

(New) Table S3: Self-reported race and ethnicity in our cohort among individuals with > 0.9 estimated native Peruvian genetic ancestry ($N = 985$)

Self-reported race	Count
American Indian + White	762
American Indian	175
White	39
Black	1
American Indian + Black	1
Asian	3
Black + White	1
American Indian + White + Asian	0
NA	3
Self-reported ethnicity	Count
Latino	979
Not Latino	4
NA	2

(New) Table S6: Association between self-reported race and TB progression. No category of self-reported race was significantly associated with TB progression ($p < 0.5$).

Self-reported race	Odds ratio (CI)	p-value
American Indian	0.94 (0.72 – 1.22)	0.63
AMERICAN INDIAN + ASIAN + WHITE	1.36 (0.51 – 3.66)	0.54
American Indian + Black	0.80 (0.56 – 1.15)	0.22
American Indian + White	0.92 (0.71 – 1.21)	0.56
Asian	1.15 (0.72 – 1.84)	0.56
Black	0.89 (0.65 – 1.20)	0.43
Black + White	0.83 (0.52 – 1.32)	0.42
White	0.91 (0.7 – 1.20)	0.51

(New) Table S7: Accounting for self-reported race. Accounting for self-reported race does not change the association between native Peruvian genetic ancestry and TB progression risk. SES: socioeconomic status; NAT: native Peruvian genetic ancestry; race: self-reported race

Model	NAT OR (CI)	NAT p-value
TB status ~ age + sex + NAT + SES + (1 HH) + GRM	1.25 (1.18-1.33)	1.1×10^{-13}
TB status ~ age + sex + NAT + SES + self-reported race + (1 HH) + GRM	1.26 (1.18-1.33)	2.3×10^{-13}

We have now updated the manuscript's text and added a new results section to describe the above results as follows:

Correlation between self-reported race and genetic ancestry

Self-reported race or ethnicity is frequently used in epidemiological or medical studies to account for an individual's background. However, self-reported race/ethnicity can be a poor proxy for genetic ancestry in admixed populations (Kumar et al., 2010; Mersha and Abebe, 2015; Sucheston et al., 2012). In our cohort, the majority of participants self-identify their race as “American Indian + White” and their ethnicity as “Latino” (74% and 99% respectively, Table 2). Genetic ancestry proportions differ significantly between self-reported race categories (ANOVA p-value $< 10^{-30}$ for all four tested genetic ancestries, Figure 2). For example, individuals who self-identified as “Black” had a higher proportion of West African genetic ancestry than the average of all other categories (mean (standard deviation) = 0.25

(0.23) vs. 0.3 (0.6), Figure 2). Nonetheless, native Peruvian ancestry was the dominant genetic ancestry in all categories of self-reported race (Figure 2). Similarly, 18% of individuals with high (> 0.9) proportion of native Peruvian genetic ancestry (N = 985) self-identified as “American Indian” compared to only 7% of individuals with low (< 0.5) proportion of native Peruvian genetic ancestry (N = 140, Table S3 and Table S4). In all tertile of native Peruvian ancestry the majority of individuals self-reported as “American Indian + White” followed by “American Indian” (Table S5). Altogether, these results suggest that self-reported race and genetic ancestry are correlated; however, individuals who self-report the same race can have drastically different levels of genetic ancestry proportions.

We then tested whether self-reported race is associated with TB progression risk. No category of self-reported race was significantly associated with TB status, suggesting that in our cohort self-reported race is not a risk factor for TB progression (Table S6).

We also added the above results about the effect of self-reported race on the association between native Peruvian ancestry and TB progression risk to the results section under “Association between genetic ancestry and TB progression risk”:

Adjusting for self-reported race (Table S7) or removing 430 related individuals (kinship coefficient ≥ 0.125) did not change these results (Figure S6, Table S8).

Finally, we updated the methods section as follows:

Testing the association between self-reported race and TB progression risk To test the association between self-reported race and TB progression we used the following logistic regression model framework implemented in basic R glm function:

equation 2:

$$\log \left[\frac{y_i}{1 - y_i} \right] = \theta + \beta_{self-reported\ race} X_{d,self-reported\ race}$$

Y_i is the probability of individual i being a case, θ is the intercept, $\beta_{self-reported\ race}$ is a vector of effect estimates for self-reported race categories (categorical variable with eight categories).

2- In the results of Table 1, it appears that the indigenous ancestry difference between cases and controls only occurs in tertiles 2 and 3 of indigenous ancestry and not tertile 1. Tertiles 2 and 3 have high indigenous ancestry. How do those individuals self-report?

Answer: The reviewer is correct, however, this observation is in line with our hypothesis that with an increasing proportion of native Peruvian ancestry (i.e. from tertile 1 to tertile 3) people are more likely to progress to active TB. In line with this, we also see the proportion of cases who self-identify as “American Indian” or “American Indian + White” is higher in tertiles 2 and 3 compared to tertile 1 (Table S5 below).

(New) Table S5: Self-reported race in native Peruvian ancestry tertiles for cases and controls.

NAT tertile	Self-reported race	Total	TB case	control
1	American Indian + White	800	410	390

	American Indian	151	75	76
	White	131	78	61
	Black	36	20	16
	American Indian + Black	5	1	4
	Asian	1	1	0
	Black + White	4	1	3
	American Indian + White + Asian	1	1	0
	NA	5	3	2
2	American Indian + White	865	553	312
	American Indian	156	111	45
	White	99	61	38
	Black	5	3	2
	American Indian + Black	9	6	3
	Asian	2	1	1
	Black + White	1	1	0
	American Indian + White + Asian	0	0	0
	NA	5	3	2
3	American Indian + White	873	596	227
	American Indian	208	137	71
	White	51	35	16
	Black	1	1	0
	American Indian + Black	1	0	1
	Asian	3	3	0
	Black + White	1	1	0
	American Indian + White + Asian	0	0	0
	NA	3	3	0

We added the above results to the manuscript text under “Correlation between selfreported race and genetic ancestry” as follows:

In all tertile of native Peruvian ancestry the majority of individuals self-reported as “American Indian + White” followed by “American Indian” (Table S5).

3- On a related topic, the authors note geographic correlates with genetic ancestry. Does that geography therefore also reflect differences in racial self-reporting?

Answer: We thank the reviewer for this comment. We now added a new figure (Figure S2B below) to show the geographical distribution of TB cases and TST positive HHCs based on the self-reported race. We do not observe any obvious geographical clustering of cases or HHCs based on self-reported race.

B

Figure S2: Geographical distribution of TB cases and TST positive HHCs B) Geographical location of TB cases and their TST negative HHC (controls) based on the most common categories of self-reported race: “American Indian + White”, “American Indian”, “White”. We reference this new figure in the manuscript:

We recruited participants from 106 district health centers in Lima, Peru in a catchment area encompassing 12 of the 43 districts of metropolitan Lima and 3.3 million inhabitants (Figure S2 A & B).

4- Because TB is an infectious disease, it is important to characterize and control for infection in considering host characteristics. To this end, the authors have chosen a design to select controls from the same households that active TB cases derive from. As the authors state, choosing controls from households controls for possible confounders. That might be true, but depends completely on how the controls are chosen. This is not a matched design and not analyzed as such. It is clear that it is not one household control for each case, especially because there are many more cases than controls. So there is really no way to know or conclude that the controls are matched for confounders.

Answer: We thank the reviewer for this comment. We agree with the reviewer that our study is not a “matched case-control study” per se. In fact, throughout the manuscript text and methods we never refer to our study design as such, we always mention that we used a “household contact study”.

For our study, the most important variable for matching was TB exposure and infection. For this reason, we selected controls from the same households as cases to ensure they have been highly exposed to M.tb and included only those household contacts with a positive TST result and no active TB, history of active TB, or previous history of positive TST. Selecting individuals from the same household allows us to better control for environmental factors related to TB risk such as socioeconomic status or unseen factors such as eating habits. We use our statistical modeling to adjust for other possible confounders such as age and sex through multivariate modeling not by matching cases and controls: “To examine the relationship between genetic ancestry and TB progression risk in Peruvians, we applied logistic regression to test the effect of the estimated fraction of native Peruvian, European, West African, and East Asian genetic ancestries on case-control status after adjusting for age, sex, and socioeconomic status.”

Please also see answers to reviewer 1 comment #1 and reviewer 2 comment #5 for more information on cases and controls.

5- Furthermore, among the controls some active cases of TB are identified. These are considered secondary cases and included among the cases for most analyses, except for one that compares these secondary cases to baseline cases, where no difference in ancestry is discovered. The authors argue that the controls are matched to the cases on exposure. However, I can't see how that is the case, because it appears they are suggesting that the "controls" were exposed at home while the cases were presumably exposed elsewhere. This also pertains to the analysis that compared a small number of baseline cases to secondary cases from the same households. We have no way to know that the exposure and length of exposure was comparable at all.

Answer: We agree with the reviewer that we cannot know if the exposure and its length was comparable for all cases or between cases and controls. For precisely this reason, by comparing secondary and primary cases we wanted to test if differences in exposure may be responsible for differences in genetic ancestry, we did not find a difference between native Peruvian ancestry proportion of secondary and primary cases suggesting that exposure differences are unlikely to drive our results.

Following the reviewer's suggestion and to avoid confusion we updated the methods to make sure we don't imply that cases and controls were matched on exposure as follows:

We determined the infection status of HHCs using TST and evaluated them for signs and symptoms of pulmonary and extrapulmonary TB disease at baseline, and at six, and 12 months after enrollment (Figure 1). Controls with a history of active TB or previous positive TST were excluded. We chose this study design because HHCs of individuals with TB are highly exposed to M.tb and are at a high risk of developing TB (Martinez et al., 2017; Reichler et al., 2018); hence our strategy allows us to focus on TB progression by including controls that were exposed to the pathogen and were infected.

6- One way to gain some insight about household exposure is to compare the TB strain between the cases and their household controls. These data are not directly provided, but in one analysis that matched for strain, the numbers of cases and controls was very small, so one might wonder whether the others were not matched and hence had different exposures.

Answer: We cannot perform the analysis suggested by the reviewer because mycobacterial interspersed repetitive unit (MIRU) genotyping can only be obtained from TB cases where M.tb can be cultured from their sputum. Individuals with latent TB do not have M.tb in their sputum.

7- Little description is given of who the household controls are. It is typical that biological family members occupy the same household. The authors do comment on utilizing a genetic correlation matrix in the analysis, but they do not report the relationship status overall between the cases and controls. For example, how many parent-child pairs are there? How many sib pairs? How many spouse pairs? And who are the others that don't share one of these relationships? Of note, sibs are virtually identical for genetic ancestry; parents and children possibly not, but tend to be similar.

Answer: Our study is not ideal to answer some of these questions. When we selected cases and controls for genotyping we intentionally biased our selection for unrelated or distantly related individuals in the same household (e.g. a spouse was preferred to sibling). We used PC-relate (Conomos et al., 2016) to estimate the pairwise kinship coefficient between 3,435 individual in our cohort. In total, 551 pairs has kinship coefficients ≥ 0.125 corresponding to second-degree relatives or closer (Figure S6 below). Of these, 36 pairs were parent-child, 72 were sib pairs, and 453 were second-degree relatives. As shown in Figure S6 below the majority of individuals in our cohort are genetically unrelated.

Please also see the answers to Reviewer 1 comment #2 where we show that removing related individuals doesn't change our association results and Reviewer 2 comment #6 where we give a more detailed explanation about our cases and controls.

(New) Figure S6: PC-relate pairwise kinship coefficients. Of the total 551 related pairs, 36 pairs were parent-child (red dashed line), 72 were sib pairs (green dashed line), and 453 pairs were second-degree relatives (blue dashed line). The rest were more distant relatives (yellow dashed line) or unrelated (black dashed line).

dashed line). Y: PC-related pairwise kinship coefficients. X: IBD sharing probability that the individuals share 0 copies of the alleles.

We updated the manuscript text to include this result:

...removing 430 related individuals (kinship coefficient ≥ 0.125) did not change these results (Figure S6, Table S6, Table S7).

8- The authors end up with about 22K markers for their analysis. It appears these markers are derived from an array that was derived from exome sequencing of indigenous Peruvians. While this number may be adequate for global ancestry estimation, it is far too sparse for admixture mapping - so it is not clear what the real power was of the admixture mapping analysis that was done. In any event, there were no significant results from the admixture mapping. The authors however, chose to highlight a locus on chromosome 5 that was not significant.

Answer: We thank the reviewer for pointing out this confusing point. To clarify, for local ancestry inference and admixture mapping we used a total of 889,203 markers after QC and pruning. We agree with the reviewer that while 22K markers can be reliably used for global ancestry inference they are unlikely to be sufficient for local ancestry inference and admixture mapping.

To make the number of SNPs used for admixture mapping clearer we updated the manuscript's text in the result section as follows:

We performed local ancestry inference followed by admixture mapping to look for specific genomic regions that might explain the association between native Peruvian genetic ancestry and TB progression risk (N = 889,203 markers following imputation, QC, and pruning).

We also updated the methods section to reflect more details about local ancestry inference and to include the number of markers used for admixture mapping as follows:

We performed local ancestry inference using PCAdmix, using imputed data to increase the number of shared markers between our data and reference data. Following phasing and imputation as described previously (Luo et al., 2019), we excluded SNPs with imputation quality score $r^2 < 0.4$, HWE p-value $< 10^{-5}$ in controls, or a missing rate per SNP greater than 5% which left 7,756,401 markers. To increase the number of overlapping variants between our cohort and the reference panel, we chose reference individuals with whole-genome sequencing data available including 25 native American individuals from the Simons Genome diversity project (Mallick et al., 2016) plus 5 individuals from the 1000 Genomes Project (Abecasis et al., 2012) PEL that had inferred native Peruvian genetic ancestry > 0.95 based on ADMIXTURE analysis at K = 4 clusters as proxy for native Peruvian genetic ancestry, 30 randomly selected individuals from the 1000 Genomes study CEU as proxy for European genetic ancestry, 30 randomly selected individuals from the 1000 Genomes Study YRI as proxy for West African genetic ancestry, and 30 randomly selected individuals from as proxy for European genetic ancestry CHS populations as proxy East Asian genetic ancestry. We then merged our data with the reference panel data and restricted the merged dataset to variants with MAF $> 5\%$ in each of the reference populations. The post-QC dataset included 2,910,169 variants. We phased the merged data using SHAPEIT2 (O'Connell et al., 2014). After phasing and following PCAdmix developer's recommendation, we used PLINK (version 1.90b3w) to remove the markers with $r^2 > 0.8$ with any other marker within a sliding window of 20 markers per window and an offset of 10 using. 889,203 variants after pruning remained for use in local ancestry inference.

Please also see the answers to Reviewer 1 comments #4 and #5 where we provide additional information on the effect of including more markers for global ancestry inference (Reviewer 1 comment #4) and the design of our genotyping array and the 712,200 markers included in this array (Reviewer 1 comment #5).

9- The authors have previously reported a genome wide association study of the same subjects, with no results in this region in the admixture mapping. There was a single locus identified in the GWAS on chromosome 3, but there is no signal here. There appears to be an allele frequency difference between indigenous Americans (low frequency) and Europeans (about 5-6%), but perhaps that's not a big enough difference for a signal (although the other paper reports that progression to TB in this population is driven by the locus on chromosome 3).

Answer: In the Luo et al paper we report a genome-wide association signal at 3q23. This signal is unlikely to explain the association between native Peruvian ancestry and TB progression risk. For one, the odds ratio in this locus is much smaller than the odds ratio we observe for the genome-wide effect of native Peruvian ancestry (1.18 for GWAS locus vs 9.72 for 0% vs 100% native Peruvian genetic ancestry). For another, the risk-increasing variant at is slightly more prevalent in Europeans than Peruvians as noticed by the reviewer, and thus unlikely to be driving the admixture mapping signal. We believe that our GWAS and admixture mapping results are complementary for understanding the genetic basis of TB progression and are pointing to two different TB susceptibility loci in the Peruvian population. Similarly, previous admixture mapping and GWAS of asthma have shown that the results can be complementary and do not always point to the same susceptibility loci (Daya et al., 2019; Galanter et al., 2014).

One conclusion that is shared between our admixture mapping and GWAS results is the polygenic architecture of TB progression risk. In our global ancestry modeling, we observe a strong genome-wide effect for the role of native Peruvian ancestry in TB progression risk relative to other tested ancestries; yet we did not find any single loci that can explain this effect in our admixture mapping. Similarly, in our GWAS we estimated the genetic heritability of TB progression in Peruvians to be 21.2%; yet we found only one genome-wide significant locus associated with TB progression risk. Both results support the hypothesis that TB progression follows a complex genetic architecture with many loci with small effect sizes involved and suggest that better-powered analyses with larger sample sizes, better phenotyping, or deeper sequencing can help identify more TB susceptibility loci including those that are ancestry-specific.

We added the following text to the discussion section to reflect the above points:

... This conclusion is in line with our previous genome-wide association study (GWAS) of TB progression in the current cohort where we showed SNP heritability (h^2_g) of TB progression in Peruvians to be 21.2%; yet found only one genome-wide significant locus (3q23) associated with TB progression risk (Luo et al., 2019). We did not identify any association in 3q23 locus in our admixture mapping analysis. However, GWAS and admixture mapping results can be complementary and do not necessarily always point to the same risk loci (Daya et al., 2019; Galanter et al., 2014).

10- The authors have included a sizeable number of covariates in their analysis, which is a plus. However, they never report whether these covariates are associated with TB progression and their significance. Of possible concern, the authors report no correlation of their measure of SES (based on home features) with genetic ancestry, while education level is inversely correlated with indigenous ancestry. It is not clear why these are not tracking similarly.

Answer: The reviewer's conventional wisdom here is that indigenous ancestry should be associated with less education and worse socioeconomic status. While this might be true in general, we note that in our cohort we recruited cases and controls from households with TB. This is a really important consideration

since living in a household with TB itself is a marker of profound socioeconomic differences. So we think that this is a major factor in our recruitment strategies that results in associations that are not always consistent with conventional wisdom. Indeed, the socioeconomic differences and educational differences between the different tertiles are less than one might expect. SES has no correlation, and although education category and native Peruvian ancestry are associated the amount of variance explained in education by native Peruvian ancestry is small (McFadden's $R^2 = 0.007$), suggesting a minimal effect of native Peruvian genetic ancestry on education in our cohort of individuals from TB households.

11- For the admixture analysis, it appears to be an unsupervised analysis that includes a range of indigenous American populations, not just Peruvians. Yet the authors report results as Peruvian indigenous ancestry rather than Native American ancestry. It appears that many different indigenous groups were used in the estimation, so this may be a bit misleading.

Answer: It is correct that with increasing the number of clusters in global admixture inference we can see more refined levels of native ancestry within native Peruvian ancestry (Figure S4 of the manuscript). However, we chose to refer to the collective proportion of native American genetic ancestry within Peruvians as native Peruvian genetic ancestry to avoid implying that our results can be generalized to all native American populations.

Different native groups in America have different demographic histories and native American genetic ancestry can vary largely from one native population to another (Posth et al., 2018; Reich et al., 2012). Moreover, the role of genetic ancestry in TB progression risk is dependent on specific populations' demography, culture, and health patterns (Moreno- Estrada et al., 2014; Spillane et al., 2020) which again varies drastically between different native American groups. For these reasons and after carefully consulting with our Peruvian and science historian co-authors we decided to refer to native American proportion of the genome of native Peruvian genetic ancestry.

12- At one point, the authors state the following: "Second, we restricted the analysis to 2043 microbiologically confirmed TB cases and 950 HHCs who were TST positive at baseline and did not develop active TB during the one year follow up and tested the association of native Peruvian genetic ancestry and TB progression risk using equation 1. Third, we restricted the analysis to 2043 microbiologically confirmed TB cases and their 950 HHCs to test the association of native Peruvian genetic ancestry and TB progression risk using equation 1."

I don't see a difference between what is second and third above. Can you clarify?

Answer: This was a mistake and we apologize for the confusion it caused. We have updated methods as follows to correct this mistake:

Second, we restricted the analysis to 2043 microbiologically confirmed TB cases and 950 HHCs who were TST positive at baseline and did not develop active TB during the one-year follow-up and tested the association of native Peruvian genetic ancestry and TB progression risk using equation 1. Third, we restricted the analysis to secondary cases (N=213) and their HHCs (N=214) and tested the association of native Peruvian genetic ancestry and TB progression risk using equation 1. Finally, we performed a sensitivity analysis using 58 secondary clustered TB cases and their 48 HHCs and tested the association of native Peruvian genetic ancestry and TB progression risk using equation 1.

13- Table 1:

Not clear why you have provided P-values for ancestry when the tertiles were defined in terms of ancestry.

Answer: It is true that tertiles are defined based on native Peruvian genetic ancestry. In Table 1 we test the association of these tertiles with each covariate that is included in our analyses. In order to have all cohort's demographic information in one place, we chose to provide pvalue for native Peruvian genetic ancestry and European genetic ancestry as well as all other covariates.

14- Age is significant, but U-shaped. That would suggest you might use age-squared in the regression analysis also.

Answer: Following reviewer's suggestion, we repeated the analysis using the following model:

TB progression ~ NAT + (age)² + sex + SES + (1|HH) + GRM

The effect of native Peruvian ancestry on TB progression risk was similar to the model with age (z=0.08, p=0.47, Response letter Table 3 below).

Response letter Table 3: Accounting age squared. SES: socioeconomic status; NAT: native Peruvian genetic ancestry.

Model	NAT OR (CI)	NAT p-value
TB status ~ age + sex + NAT + SES + (1 HH) + GRM	1.25 (1.18-1.33)	1.1x10 ⁻¹³
TB status ~ age ² + sex + NAT + SES + (1 HH) + GRM	1.26 (1.18-1.33)	5.4x10 ⁻¹⁴

We did not add the above table to the manuscript because the analysis results show that including age instead of age-squared does not change our results.

15- Females appear to have much more indigenous ancestry than the males. Why is that the case? And how does it relate to case-control status? Did you stratify your analyses by sex?

Answer: Our cohort includes 1924 males (1296 cases and 628 controls) and 1501 females (802 cases and 692 controls). It is true that in our cohort females seem to have slightly higher native Peruvian genetic ancestry than males (0.82 vs 0.79, t-test two-sided p=4.5x10⁻¹⁰, Figure S7 below). This is true for both cases and controls (for cases 0.84 vs. 0.81, p=2.7x10⁻⁶ and for controls 0.80 vs. 0.75, p=1.8x10⁻⁹). We cannot comment on why that is but it probably reflects the demography of our cohort and not the whole population. Following the reviewer's suggestion and in order to test if the effect of native Peruvian genetic ancestry differs between the two sex, we stratified the cohort by sex and repeated the analysis using the following model:

TB progression ~ NAT + age + SES + (1|HH) + GRM

The effect of native Peruvian ancestry on TB progression risk was not significantly different between the two sex (overlapping confidence intervals, z=0.46, p=0.32, Table S9 below).

(New) Figure S7: Proportion of native Peruvian genetic ancestry in males and females. Females have slightly higher native Peruvian genetic ancestry than males (0.82 vs 0.79, t -test two sided $p=4.5 \times 10^{-10}$).

(New) Table S9: Sex-stratified analysis.

Model	NAT OR (CI)	NAT p-value
Male only	1.25 (1.15-1.36)	1.4×10^{-7}
Female only	1.29 (1.16-1.43)	1.0×10^{-6}

We

We added the above results to the manuscript:

Similarly, stratifying by sex did not change our results (Figure S7, Table S9).

16- The authors report the following analysis: "We considered that cases and controls who were from the same household may share a more similar ancestry profile compared to average which may bias our results. For this, we tested the association of native Peruvian genetic ancestry with TB progression risk using half of the cases ($N=791$) and the same number of controls who were not from the same household as cases and after correction for age, sex, socioeconomic status, and genetic relatedness. This analysis had a similar result to the analysis performed using the whole cohort (ORNAT0.1=1.19 (1.10-1.28), p -value= 1.0×10^{-12})."

Including cases and controls that are matched on ancestry would have attenuated the OR. But here, it appears that the opposite is the case. Unclear why.

Answer: To clarify we did not match cases and controls on ancestry in any of our analyses. We wanted to test if being from the same household can have an effect on our results, considering the ancestry of people in the same household may be more similar than people who don't share the same household. Our results from the above analysis showed that this is not the case. We did not observe a significant difference between native Peruvian ancestry odds ratio for our primary analysis and the above analysis (1.25 (1.15-1.36) vs 1.19 (1.10- 1.28), overlapping confidence intervals, $z=1.2$, $p=0.1$). It is difficult to draw conclusions about the odds ratio since they are not significantly different from each other.

17- The authors state the following in their discussion: "Our results suggest that clinicians and public health authorities in Peru should pay special attention to TB risk in areas where a large portion of the population are known to have higher levels of native Peruvian ancestry in order to ensure that the increased risk of TB progression in these individuals does not lead to increased health disparities."

What kind of special attention? Don't they already know those locations have high risk, irrespective of ancestry? Or are you suggesting that they test people's genetic ancestry in those locations and only attend to those with highest indigenous ancestry? Which health disparities are you talking about? Usually health disparities are defined in terms of race and/or ethnicity. Are you talking about the higher risk of TB among indigenous Peruvians? This really requires clarification.

Answer: We agree that our comments might be misconstrued. To avoid concerns similar to the one raised by the reviewer or avoiding the interpretation that we suggest medical care should only be given to those with a certain ancestry, or upon testing for genetic ancestry, we removed the sentences quoted by the reviewer from the discussion.

18- In a related passage, the authors state: "the importance of conducting large-scale host genetic studies in diverse populations in order to reduce health disparities and to get a comprehensive picture of the genotype-phenotype relationship in TB and other infectious diseases." Again, the authors should clarify how the health disparity is defined, and precisely how this study will help reduce it.

Answer: The majority of human genomics studies are done in populations of European ancestry (Sirugo et al., 2019). This European bias has led to misdiagnosis of Mendelian disease such as hypertrophic cardiomyopathy in non-European populations (Manrai et al., 2016). Similarly, for complex traits, this lack of diversity had led to imprecise estimation of genetic risk in non-Europeans (Martin et al., 2017, 2019) as well as missing population-specific variants with large effects on complex traits (Asgari et al., 2020). As human genomics of complex traits becomes more and more applied in the clinical setting (Sun et al., 2021) lack of representation from diverse human populations can exacerbate existing health inequalities and limit the clinical benefits of human genomics in populations of non- European ancestry. Studies like ours underlie these important points by showing that genetic risk for complex diseases can differ across different populations and thus to get an accurate view of the genetic risk and the underlying genetic variants we have to study all human populations.

We made these points clearer by updating the discussion as follows:

Our results also highlight that differences in infectious disease burden among different populations cannot be solely attributed to variations in sociodemographic factors and can be partially due to genetic differences. Currently, the majority of human genomics studies of complex traits are done in populations of European ancestry (Sirugo et al., 2019). However, with the increasing clinical applications of complex trait genomics data (Sun et al., 2021), this European bias can lead to increased health disparities (Martin et al., 2019). Our results underline the importance of conducting large-scale human genomics studies in diverse populations in order to get a better understanding of population-specific genetic risk for TB and

other complex diseases, to get a comprehensive picture of the genotype-phenotype relationship, and to enable all human populations benefit from the results of human genomics research.

20- Finally, I have a concern about novelty of this presentation. In the authors' previous GWAS paper published in Nature Communications, it appears that the same admixture figure, Figure 2a in the current manuscript, was previously published there in Figure 2a (or at least a pretty close facsimile thereof - one can easily see the admixture differences between cases and controls in the figure previously published, as is also reported here).

Answer: We have previously published a TB GWAS (Luo et al., 2019) using this cohort and as the reviewer points out have provided an admixture plot for TB cases and controls in the paper. The Luo et al paper, however, does not explore or discuss the effect of genetic ancestry on TB progression risk. In our paper, we go to a great extent to explore this question and to test whether the resulting association can be driven by non-genetic factors. This required substantial consideration of covariates and confounders. Indeed in Luo et al, ancestry is treated as a simple confounder (as it often is in GWAS studies). We believe the two studies are complementary for understanding the genetic basis of TB progression for the following reasons:

Besides our GWAS paper, the role of indigenous ancestry in the apparently increased burden of TB among native populations of America has been extensively debated for over 200 years (Jones, 2003; McMillen, 2008). However, our study is different from these previous genetic studies in three ways:

- 1- Our study uses genetic data to quantitatively assign genetic ancestry whereas previous studies used self-reported ancestry which is often a poor proxy for genetic ancestry in admixed populations (Kumar et al., 2010; Mersha and Abebe, 2015) and thus can lead to misclassification of participants.
- 2- We carefully phenotyped all individuals and ascertained infection status using TST to ensure that all individuals are exposed to M. tb. This is an important distinction as different genetic factors might underlie different stages of the disease (e.g. infection versus progression upon infection) (Luo et al., 2019).
- 3- Our household contact, longitudinal study design allowed us to ensure that all controls were exposed to M.tb and to rigorously account for potential non-genetic factors that track with genetic ancestry.
- 4- To our knowledge, this study is the first large-scale, unbiased genetic study to look at the effect of indigenous ancestry on TB or TB progression risk in South or Latin American populations.

We made the above points clearer in the discussion as follows:

... We did not identify any association in 3q23 locus in our admixture mapping analysis. However, GWAS and admixture mapping results can be complementary and do not necessarily always point to the same risk loci (Daya et al., 2019; Galanter et al., 2014).

... To our knowledge, this study is the first large-scale, unbiased genetic study to look at the effect of indigenous ancestry on TB or TB progression risk in South or Latin American populations.

Our study is different from these previous genetic studies in three ways. First, our study uses genetic data to quantitatively assign genetic ancestry whereas previous studies used self-reported ancestry which is often a poor proxy for genetic ancestry in admixed populations (Kumar et al., 2010; Mersha and Abebe, 2015) and thus can lead to misclassification of participants. Second, we carefully phenotyped all individuals and ascertained infection status using TST to ensure that all individuals are exposed to M. tb. This is an important distinction as different genetic factors might underlie different stages of the disease (e.g. infection versus progression upon infection) (Luo et al., 2019). Third, our household contact,

longitudinal study design allowed us to ensure that all controls were exposed to M.tb and to rigorously account for potential non-genetic factors that track with genetic ancestry.

References:

- Abecasis, G.R., Auton, A., Brooks, L.D., A, D.M., Durbin, R.M., Handsaker, R.E., Kang, H.M., Marth, G.T., and A, M.G. (2012). An integrated map of genetic variation from 1,092 human genomes. *Nature* 491, 56–65.
- Alexander, D., Novembre, J., and Lange, K. (2009). Fast model-based estimation of ancestry in unrelated individuals. *Biotechfor* 19, 1655–1664.
- Asgari, S., Luo, Y., Akbari, A., Belbin, G.M., Li, X., Harris, D.N., Selig, M., Bartell, E., Calderon, R., Slowikowski, K., et al. (2020). A positively selected FBN1 missense variant reduces height in Peruvian individuals. *Nature* 582, 234–239.
- Chimusa, E.R., Zaitlen, N., Daya, M., Möller, M., van Helden, P.D., Mulder, N.J., Price, A.L., and Hoal, E.G. (2014). Genome-wide association study of ancestry-specific TB risk in the South African Coloured population. *Hum. Mol. Genet.* 23, 796–809.
- Comas, I., Coscolla, M., Luo, T., Borrell, S., Holt, K.E., Kato-Maeda, M., Parkhill, J., Malla, B., Berg, S., Thwaites, G., et al. (2013). Out-of-Africa migration and Neolithic coexpansion of *Mycobacterium tuberculosis* with modern humans. *Nat. Genet.* 45, 1176–1182.
- Conomos, M.P., Reiner, A.P., Weir, B.S., and Thornton, T.A. (2016). Model-free Estimation of Recent Genetic Relatedness. *Am. J. Hum. Genet.* 98, 127–148.
- Consortium, 1000, Auton, A., Brooks, L.D., Durbin, R.M., Garrison, E.P., Kang, H.M., Korbel, J.O., Marchini, J.L., Shane, M., A, M.G., et al. (2015). A global reference for human genetic variation. *Nature* 526, 68–74.
- Daya, M., Rafaels, N., Brunetti, T.M., Chavan, S., Levin, A.M., Shetty, A., Gignoux, C.R., Boorgula, M.P., Wojcik, G., Campbell, M., et al. (2019). Association study in African-admixed populations across the Americas recapitulates asthma risk loci in non-African populations. *Nat. Commun.* 10, 880.
- Galanter, J.M., Fernandez-Lopez, J.C., Gignoux, C.R., Barnholtz-Sloan, J., Fernandez-Rozadilla, C., Via, M., Hidalgo-Miranda, A., Contreras, A.V., Figueroa, L.U., Raska, P., et al. (2012). Development of a panel of genome-wide ancestry informative markers to study admixture throughout the Americas. *PLoS Genet.* 8, e1002554.
- Galanter, J.M., Gignoux, C.R., Torgerson, D.G., Roth, L.A., Eng, C., Oh, S.S., Nguyen, E.A., Drake, K.A., Huntsman, S., Hu, D., et al. (2014). Genome-wide association study and admixture mapping identify different asthma-associated loci in Latinos: the Genes-environments & Admixture in Latino Americans study. *J. Allergy Clin. Immunol.* 134, 295–305.
- Gravel, S. (2012). Population genetics models of local ancestry. *Genetics* 191, 607–619.
- Jones, D.S. (2003). Virgin Soils Revisited. *William Mary Q.* 60
- Kerner, G., Laval, G., Patin, E., Boisson-Dupuis, S., Abel, L., Casanova, J.-L., and Quintana-Murci, L. (2021). Human ancient DNA analyses reveal the high burden of tuberculosis in Europeans over the last 2,000 years. *Am. J. Hum. Genet.* 108, 517–524.
- Kumar, R., Seibold, M.A., Aldrich, M.C., Williams, L.K., Reiner, A.P., Colangelo, L., Galanter, J., Gignoux, C., Hu, D., Sen, S., et al. (2010). Genetic ancestry in lung-function predictions. *N. Engl. J. Med.* 363, 321–330.
- Luo, Y., Suliman, S., Asgari, S., Amariuta, T., Baglaenko, Y., Martínez-Bonet, M., Ishigaki, K., Gutierrez-

Arcelus, M., Calderon, R., Lecca, L., et al. (2019). Early progression to active tuberculosis is a highly heritable trait driven by 3q23 in Peruvians. *Nat. Commun.*, 3765.

Mallick, S., Li, H., Lipson, M., Mathieson, I., Gymrek, M., Racimo, F., Zhao, M., Chennagiri, N., Nordenfelt, S., Tandon, A., et al. (2016). The Simons Genome Diversity Project: 300 genomes from 142 diverse populations. *Nature* 538, 201–206.

Manrai, A.K., Funke, B.H., Rehm, H.L., Olesen, M.S., Maron, B.A., Szolovits, P., Margulies, D.M., Loscalzo, J., and Kohane, I.S. (2016). Genetic Misdiagnoses and the Potential for Health Disparities. *N. Engl. J. Med.* 375, 655–665.

Martin, A.R., Gignoux, C.R., Walters, R.K., Wojcik, G.L., Neale, B.M., Gravel, S., Daly, M.J., Bustamante, C.D., and Kenny, E.E. (2017). Human Demographic History Impacts Genetic Risk Prediction across Diverse Populations. *Am. J. Hum. Genet.* 100, 635–649.

Martin, A.R., Kanai, M., Kamatani, Y., Okada, Y., Neale, B.M., and Daly, M.J. (2019). Clinical use of current polygenic risk scores may exacerbate health disparities. *Nat. Genet.* 51, 584–591.

Martinez, L., Shen, Y., Mupere, E., Kizza, A., Hill, P.C., and Whalen, C.C. (2017). Transmission of Mycobacterium Tuberculosis in Households and the Community: A Systematic Review and Meta-Analysis. *Am. J. Epidemiol.* 185, 1327–1339.

McMillen, C.W. (2008). “The red man and the white plague”: rethinking race, tuberculosis, and American Indians, ca. 1890-1950. *Bull. Hist. Med.* 82, 608–645.

Mersha, T.B., and Abebe, T. (2015). Self-reported race/ethnicity in the age of genomic research: its potential impact on understanding health disparities. *Hum. Genomics* 9, 1.

Moreno-Estrada, A., Gignoux, C.R., Fernández-López, J.C., Zakharia, F., Sikora, M., Contreras, A.V., Acuña-Alonzo, V., Sandoval, K., Eng, C., Romero-Hidalgo, S., et al. (2014). Human genetics. The genetics of Mexico recapitulates Native American substructure and affects biomedical traits. *Science*, 1280–1285.

O’Connell, J., Gurdasani, D., Delaneau, O., Pirastu, N., Ulivi, S., Cocca, M., Traglia, M., Huang, J., Huffman, J.E., Rudan, I., et al. (2014). A general approach for haplotype phasing across the full spectrum of relatedness. *PLoS Genet.* 10, e1004234.

Pereira, V., Santangelo, R., Børsting, C., Tvedebrink, T., Almeida, A.P.F., Carvalho, E.F., Morling, N., and Gusmão, L. (2020). Evaluation of the Precision of Ancestry Inferences in South American Admixed Populations. *Front. Genet.* 11, 966.

Posth, C., Nakatsuka, N., Lazaridis, I., Skoglund, P., Mallick, S., Lamnidis, T.C., Rohland, N., Nägele, K., Adamski, N., Bertolini, E., et al. (2018). Reconstructing the Deep Population History of Central and South America. *Cell* 175, 1185–1197.e22.

Price, A.L., Weale, M.E., Patterson, N., Myers, S.R., Need, A.C., Shianna, K.V., Ge, D., Rotter, J.I., Torres, E.,

Taylor, K.D., et al. (2008). Long-range LD can confound genome scans in admixed populations. *Am. J. Hum. Genet.* 83, 132–135; author reply 135–139.

Reich, D., Patterson, N., Campbell, D., Tandon, A., Mazieres, S., Ray, N., Parra, M.V., Rojas, W., Duque, C.,
Mesa, N., et al. (2012). Reconstructing Native American population history. *Nature* 488, 370–374.

Reichler, M.R., Khan, A., Sterling, T.R., Zhao, H., Moran, J., McAuley, J., Bessler, P., Mangura, B., and Tuberculosis Epidemiologic Studies Consortium Task Order 2 Team (2018). Risk and Timing of Tuberculosis Among Close Contacts of Persons with Infectious Tuberculosis. *J. Infect. Dis.* 218, 1000–1008.

Santos, H.C., Horimoto, A.V.R., Tarazona-Santos, E., Rodrigues-Soares, F., Barreto, M.L., Horta, B.L., Lima Costa, M.F., Gouveia, M.H., Machado, M., Silva, T.M., et al. (2016). A minimum set of ancestry informative markers for determining admixture proportions in a mixed American population: the Brazilian set. *Eur. J. Hum. Genet.* 24, 725–731.

Sirugo, G., Williams, S.M., and Tishkoff, S.A. (2019). The Missing Diversity in Human Genetic Studies. *Cell* 177, 26–31.

Spillane, N.S., Treloar Padovano, H., and Schick, M.R. (2020). Regional and gender differences in tobacco use among American Indian youth. *J. Ethn. Subst. Abuse* 19, 553–566.

Sucheston, L.E., Bensen, J.T., Xu, Z., Singh, P.K., Preus, L., Mohler, J.L., Su, L.J., Fontham, E.T.H., Ruiz, B., Smith, G.J., et al. (2012). Genetic ancestry, self-reported race and ethnicity in African Americans and European Americans in the PCaP cohort. *PLoS One* 7, e30950.

Sun, J., Wang, Y., Folkersen, L., Borné, Y., Amlien, I., Buil, A., Orho-Melander, M., Børghlum, A.D., Hougaard, D.M., Regeneron Genetics Center, et al. (2021). Translating polygenic risk scores for clinical use by estimating the confidence bounds of risk prediction. *Nat. Commun.* 12, 5276.

Woodman, M., Haeusler, I.L., and Grandjean, L. (2019). Tuberculosis Genetic Epidemiology: A Latin American

Referees' report, second round of review

Referee #1

No comment

Referee #2

The authors have answered all my questions/concerns

Referee #3

I appreciate the authors' responses to my previous concerns, the additional analyses performed and clarifications to the manuscript. The paper is improved now, but I still have some concerns that should be addressed that were unanswered.

1. Thank you for including all the additional information, tables, figures and analyses about self-reported race and ethnicity. These new data and analyses are very helpful for interpreting the results and comments made about them.

From all the analyses, it does appear that there is no significant difference between cases and controls in terms of self-reported race and ethnicity (for ethnicity, since nearly all subjects

identified as “Latino”, there was really nothing to analyze there). Latino is an ethnic designation in the US. Is it a designation in Peru?

It is also true that there is an indigenous ancestry difference between cases and controls within each race category. However, it does appear to attenuate in those categories that are not identified with Native American race:

	mean ancestry		
	case	control	delta
AI+white	0.825	0.780	0.045
AI	0.835	0.797	0.038
white	0.770	0.747	0.023
black	0.677	0.662	0.015

This is also consistent with the observation discussed later that the OR for TB progression may not be linear with indigenous genetic ancestry, as it seems to attenuate in the lower range of ancestry. This should be noted.

Line 219: It would be helpful to define the race categories here, and how they were determined – from a fixed list on a survey, open ended question, etc.

Line 288 The authors state “Self-reported race or ethnicity is frequently used in epidemiological or medical studies to account for an individual's background. However, self-reported race/ethnicity can be a poor proxy for genetic ancestry in admixed populations.”

As I stated before, it is not clear why the authors are saying this. Can they cite a reference claiming that race/ethnicity is a **good proxy** for genetic ancestry? I believe what is in the literature is that race/ethnicity is correlated with genetic ancestry. As I said before, they are not used synonymously. Race and ethnicity are personal identifiers. Genetic ancestry is estimated from genetic data.

Line 454-458 is clearly an overstatement, and self-contradictory. In many places the authors cite that indigenous groups have higher TB risk, but that is based on racial characterization, not on genetic ancestry. It is also completely unclear what “misclassification” here means. What classification are we talking about? It is more a matter that race differences alone do not invoke a genetic explanation for a difference. Also, the authors have said nothing here about whether they believe their genetic ancestry result explains a race difference.

All said, the most important conclusion from the analysis of this study cohort is that there is no association of self-reported race with TB progression – specifically, that those that identify as Native American are not at greater risk of clinical TB compared to those who self-identify as white. That should be clearly stated in the abstract as well as the results and discussion.

2. Linearity of relationship of indigenous genetic ancestry with OR for active TB. In the abstract, the authors comment that a 10% increase in native ancestry is associated with 25% increased risk; then say that the highest decile has 3-fold increased risk compared to lowest decile. If the risk increases 25% per decile, then shouldn't the highest decile have a 2.5-fold increased risk compared to the lowest? Also, this raises the question of possibly non-linearity of the relationship. According to Figure 3c, the lowest decile (with an average of 48% indigenous genetic ancestry) doesn't really fit the linear model well. It also has a much broader variation in ancestry compared to the other deciles. It would be interesting to further subdivide this decile to see if there is an actual decline in risk with lower indigenous ancestry within this decile. This is important in terms of extrapolating the genetic ancestry results from a linear model into this range.

3. Thank you for the map, which is helpful. In Figure S2B, the green appears much more dominant than red, but green is supposed to be AI race while red is AI+white. In the manuscript tables and text, it is the AI+white group that is dominant. Are those colors backwards? Also, there does appear to be a differential geographic distribution of green versus red in the figure which should be noted.

4. The authors state that their study was not a “matched” case-control study, but then say: “For our study, the most important variable for matching was TB exposure and infection. For this reason, we selected controls from the same households as cases to ensure they have been highly exposed to M.tb and included only those household contacts with a positive TST result and no active TB, history of active TB, or previous history of positive TST. Selecting individuals from the same household allows us to better control for environmental factors related to TB risk such as socioeconomic status or unseen factors such as eating habits. We use our statistical modeling to adjust for other possible confounders such as age and sex through multivariate modeling not by matching cases and controls.”

This seems a contradiction and a misunderstanding. Certainly they could have chosen individuals from other households as controls that had the same evidence of TB exposure and infection. As they go on to say, the matching by household still allows for control for known and measured environmental factors by including them as covariates in the logistic model (which is true). However, they also say the matching allows for control of unseen factors such as eating habits, which is untrue. You cannot control for unmeasured confounders in an unconditional logistic regression analysis. For that, you need a matched or conditional logistic regression analysis. So, the current analysis leaves open the possibility of confounding by unmeasured variables, and the authors need to say so.

Another negative is that the choice of controls has led to overmatching of the exposure between cases and controls (i.e. genetic ancestry) because individuals living in the same household tend to be related and therefore more similar for genetic ancestry than controls not from the same household.

While there are pluses to the study design and choice of controls and analysis, there are also minuses. The authors need to be more explicit about the minuses in the discussion.

There are also missing important details of the study design, as noted below.

Line 109: Index cases are defined as the first TB case in the household. When there is more than one TB case in a household, how was it determined who was first? And by first, it is the one who became symptomatic first, or who first had a positive TST? Also, if there were multiple cases within a household, did you include just one? This especially needs clarification because cases that first became symptomatic after the study enrollment were included (leading to multiple cases per household).

Line 112: controls are TB infected but not symptomatic after 1 year of follow up, and had no history of active TB.

Line 122: previous TB is a confounder, but controls were excluded if they had previous TB? So cases were not excluded if they had previous TB?

These two lines especially need clarification. If controls could not have prior TB, then prior TB cannot be considered as a confounder in any analysis. Only cases could have had prior TB.

Line 125-128: some HHCs developed TB; these are called baseline cases if within 14 days; secondary cases if after 14 days. Again, this begs the question of how many cases per household?

Line 138-139: Children were included? What was the age range of the subjects? Nutritional status was obtained only for children? If so, how could this have been analyzed?

Line 191: case control analysis by unconditional logistic regression, no matching. This matching made the controls more comparable to cases for some variables like SES which were based on a household definition (type of house). However, this also depends on the number of controls versus cases obtained from each household. To understand this better, the authors should provide a supplemental table with number of families with 1 case and no control, 1 case and 1 control, 1 case and 2 controls, 2 cases and 0 controls, 2 cases and 1 control, 2 cases and 2 controls, etc.

Line 207: Here you are saying 2043 microbiologically confirmed TB cases but above on line 182 you said there were 1929 TB cases; how can a subset be larger than this? Perhaps all analyses were based on inclusion of all related controls except for one analysis excluding them. The authors need to be clear about this and the numbers.

Line 210: I am confused here. Secondary cases had controls? I thought the secondary cases were controls that ended up developing TB? So how do they have controls? Are secondary cases included as cases in the case total, the HHC total? Same question for the 58 clustered cases – who were originally controls – how do they have controls?

Line 345: I thought that prior TB was an exclusion factor for controls, so the controls couldn't have any prior TB.

Line 351: While the authors have mentioned a number of covariates, they have not included other possible confounders. For example, they haven't included many other clinical risk factors – e.g. diabetes. Diabetes increases the risk of TB by 3x. General health status is probably an important covariate. This limitation also needs to be stated in the discussion.

Line 359: Does this mean that HHCs that developed TB were still included as controls in the original analyses? Who were the HHCs that were not included in this analysis?

Line 370-371: Secondary cases more likely to be household transmission. But how do you know which case was infected first, especially given a significant lag time before symptoms may occur? And how do you know what kind of exposure the “secondary case” had outside the household?

Line 373-374: How do secondary cases have HHCs? I thought they were the original HHCs that became positive. Same applies to “clustered cases”. Again, this relates to the question about numbers of controls and cases per household.

Line 461-463. Their study is only somewhat longitudinal and it was never analyzed as a longitudinal study but as a case control study. It has no data on time of infection, and also in no way controls for infection outside of home, for example in a work setting. These are limitations that should be mentioned.

5. Were cases with a prior history of TB not excluded? Then the ancestry could be correlated with prior history of TB and prior exposure.

Line 384: “Native Peruvian genetic ancestry proportion was similar between baseline and secondary cases from the same households which is consistent with the conclusion that differences in native Peruvian genetic ancestry proportion are not associated with differences in exposure (N=135 baseline and 213 secondary cases, ORNAT0.1=1.02 (0.86-1.22), p-value=0.28, Table S11”

It is not clear how that analysis leads to that conclusion. What you have compared is two groups of individuals that progressed to active TB, but the timing and degree of their exposure is

unknown. Presumably they were both exposed. You did not compare the ancestry of exposed and unexposed individuals, which was not feasible in this study design.

7. Thanks for including a figure with the kinship results.

Line 178: It would be nice to also include a supplementary table where the number of relative pairs are stratified as both cases, both controls, or case-control. Also, most people would consider first cousins (kinship=.06265) as related individuals, so it would be useful to include the number of such pairs also. In randomly removing individuals, was there preferential removal of controls vs cases? Ended up with almost twice as many cases as controls

Line 199: “GRM is a genetic relatedness matrix. It has values from 0-.5.” But above you said you eliminated individuals so that there are no pairs with relatedness above 0.125. It should be clarified which analyses did or did not exclude related individuals. Again, it should not be stated that the analysis removing individual with kinship above 0.125 left the remainder as unrelated.

Line 315: “cryptic relatedness” – not all the relatedness is cryptic, as you have many first, second and third degree relatives, and no doubt 4th and 5th also that are known to the subjects.

Line 320: It appears all analyses except one included related individuals as controls?

8 and 9. The additional information about how local ancestry estimation was performed and number of SNPs is welcome. However, this still requires some clarification.

Line 76: “Here we aim to understand the role of ancestry-specific genetic factors that affect TB progression... “

This statement is misleading, because the authors only found an association with overall genetic ancestry which can be correlated with non-genetic factors. Perhaps better to say “ancestry-related.”

Line 167-172: “ADMIXTURE (Alexander et al., 2009) (version 1.3) with K = 4-7 clusters to perform global ancestry inference on this dataset. We used reference populations in ADMIXTURE analysis to determine what genetic ancestry each cluster represents. For example, if a cluster was the dominant cluster in the European individuals from the 1000 Genomes Project (Consortium et al., 2015) we concluded that this cluster represents European genetic ancestry in admixed Peruvians from our cohort. All genetic analyses were done using GRCh37”

This strongly suggests that unsupervised admixture analysis was used. This is known to be inferior to using fixed reference populations. Also, it makes no sense – because specific non-admixed samples were deliberately chosen as reference for this analysis. To this point, in Figure 3A – EUR looks to have some NA and EAS genetic ancestry. This is not reasonable and is likely an artifact of the program in an unsupervised analysis.

If, in fact, this was an unsupervised analysis, the authors need to say so and also the limitations of doing so (albeit not likely to have changed the overall admixture results much).

Line 390 et seq: For the admixture mapping, you performed (at least) two tests (possibly others not reported), but only corrected for one test, and not clear this decision was a priori. In any event, these results are not significant.

Table S13. The variants in this table have modest P-values and are clearly not significant at all allowing for multiple testing, and the betas are also quite small. It is clear they cannot explain the admixture signal here, which was also not significant.

This whole section is a bit of an overreach to claim something location specific for a genetic ancestry effect. It is also clear that no previous loci identified by GWAS show an ancestry effect in the analysis here, and that should be stated. The problem is that without identified loci, the genetic ancestry effect may be non-specific and not due to directly predisposing genes and this needs to be stated, particularly in the discussion.

Line 436-440. Because there is very little East Asian ancestry in this cohort, that could be left out of the admixture mapping. In fact, you could also leave out the African, just looking at NA versus European to improve the efficiency of the local admixture program. Although it is doubtful the results would be any different since those were likely the major locus specific ancestries in the analysis that was performed.

Line 448-450. The two cited studies, one a candidate gene, the other linkage (which appeared to implicate the gene NRAMP1), are old and apparently have not been replicated. Certainly the authors could say whether their study replicates them or not. If not, they shouldn't be cited as evidence.

Line 433-435. It is well known that GWAS is generally more powerful than admixture mapping because the latter depends heavily on large ancestral allele frequency difference. The authors should simply state here that their chromosome 3 locus does not explain any ancestry difference; they could also show the allele frequencies at their top chromosome 3 SNP in the different ancestral groups.

10. This is not an issue of conventional wisdom, it is an issue of biased study design. In most places education level is associated with higher income and therefore higher SES. I imagine that is also probably the case in the general population of Peru. Here SES is determined by house materials. SES is effectively matched between cases and controls by choosing controls from the same household (group variable) but education level is individual. The inverse correlation of indigenous ancestry with education could be a marker of earlier exposure (e.g. by occupation) of cases (and also prior TB if they were not excluded for having prior TB).

11. I still have a disagreement about the use of the term Native Peruvian both in the title and throughout the paper, for a number of reasons. As reference in the admixture analysis, you did not only include indigenous people currently living in Peru, but elsewhere. Peru has only existed as an entity for a few hundred years. The indigenous people predate the creation of the country of Peru. The Inca empire stretched from Ecuador to Chile. Furthermore, this is the only reference population for the admixture analysis you characterize by a country name. For example, you use general Europeans for a reference and don't specifically use or say Spaniards. You don't say Nigerian or Yoruba but African; you don't say Chinese/Japanese but East Asian. You should be consistent and accurate in these reference population characterizations, otherwise it appears misleading and biased. So, it would be preferable to say either Indigenous, Indigenous American, or Native American.

Regarding the concern about generalization to other indigenous groups, such lack of generalization is much more likely due to social and environmental differences than genetic ancestry ones. So again it is misleading to imply that such differences among indigenous groups are genetic.

13. P-values are used to test hypotheses. What is the hypothesis here? That mean indigenous ancestry differs among tertiles that were defined based on indigenous ancestry? To avoid confusion, probably best to take out the P-values here but leave the means in.

15. It is good that the analysis stratified on sex showed similar results. However, the interpretation that the sex difference in ancestry between cases and controls is a reflection of the demography of the study sample and not the general population suggests some type of ascertainment or sampling issue, which should be mentioned. This could apply to other demographic aspects, and it should be acknowledged that the sampling and results don't necessarily extend to the general population.

16. unmatched analysis

Line 362-367: This is now an unmatched analysis. These controls also come from other households of cases. So how are they controls? Also, the OR is attenuated.

Some additional major comments

Line 330: only NAT ancestry is associated with increased risk, and the other ancestries are non-significant after including NAT ancestry. But in the introduction you said it was the Europeans who had been exposed to TB and therefore had decreased allele frequencies. That is a contradiction and would a priori suggest that model and interpretation is not correct, since it was not EUR ancestry that predominates. In fact, the entire interpretation of differential selection across different continents due to TB needs to be toned down, because the authors have no evidence to support this. Pre-Columbian South America also had prevalent TB, as did Asia and Africa. Again, because there are no results of specific genes, and the specific genes that have been previously identified by GWAS or otherwise (by these authors and others) show no ancestry effect regarding indigenous American populations, the genetic ancestry correlation alone cannot be used to support such a prior selection model.

Line 419-425. These sentences are self contradictory. With indigenous ancestry in the model, none of the others are significant including EUR. As above, the model of TB endemicity in Europe and subsequent selection is not a good model here.

Line 439-440. Apparently, a number of loci have already been identified by GWAS (the authors cite them). Instead of talking about a power issue here, it would be preferable to actually show the results of those SNPs here, or at least state whether those prior results were replicated in this study, and if they show any ancestry differences.

Line 442-443. “To our knowledge, this study is the first large-scale, unbiased genetic study to look at the effect of indigenous ancestry on TB or TB progression risk in South or Latin American populations.”

Why say unbiased? I’m not sure that is true in this case, and it seems to imply that prior studies were biased. If that authors believe so, they should name the studies and say what the biases were. If this is the first large scale study, just leave it at that. Also, here the term “indigenous ancestry” is used (which is preferable) but throughout the manuscript it says Native Peruvian. It seems that the intent here is to generalize beyond current Peru.

Line 462-463. Seems an overstatement to say “rigorously” account for factors that track with genetic ancestry. It probably matches on some that are based on cohabitation, but possibly not others. This statement should be more measured (perhaps just remove the word rigorously).

Lines 465-469: Thanks you for this appropriate note of caution in interpreting the results. However,

Lines 469-471: This statement is speculative, and undermines the appropriate caution above. Also, not sure how you “controlled” for household since that was not a covariate in the analysis. You could say SES, however, as measured by physical house characteristics.

Line 471-473. Varies across which populations? NAs versus whites? You have shown a genetic ancestry correlation, not a race difference. You haven’t defined “population” but a typical reader would likely interpret this as racial or ethnic or nationality group.

Line 473-476. This is a very important statement. And also should provide additional caution about the ancestry claims made here, that the explanation is genetic, as no specific genes have been identified.

Additional minor comments

>Abstract: Refers to TB progression. 10% increase in native ancestry associated with 25% increased risk; then say highest decile has 3-fold increased risk compared to lowest decile. But shouldn't it be 2.5-fold? Also, to avoid confusion, you should point out that the lowest decile had 48% indigenous ancestry while the highest decile had 98% (Figure 3C). Also, the 25% is based on the linear model, not on actual numbers? This assumes a linear relationship? Figure 3C suggests that the relationship may not be linear throughout its range.

Line 175: GENESIS R package (no reference is given).

Line 193: the way betas for age and sex are defined as vectors with plural, it sounds like multiple estimates – rather than one beta estimate

Line 195: you have beta-anc in the model on line 191 but here are saying beta-nat. You need to clarify this. I assume beta-anc is allowing for the different ancestries one at a time in separate analyses?

Line 196: Also you refer to those ancestries as beta-nat; I assume you mean beta-anc?

Line 205: here you are saying West African and East Asian ancestry are other covariates, but you already mentioned them above. Also, what about European?

Line 324: it says conditional analysis, but it doesn't look like that; it looks like just having both ancestries in the model at the same time; if it truly was conditional, that needs to be said; joint analysis is preferred, or you also need to condition first on the other ancestry and then test NAT

Lines 333 and 337: I think you mean figure 3b and 3c?

Line 481-485. The same statement was made twice. No doubt it is a truism that there are genes that predispose to infection and progression of an infectious disease, and there will be differences in allele frequencies across populations. How much these differences account for an observed prevalence difference between populations or race/ethnicity groups is another matter, and to this the authors have really not provided evidence.

Authors' response to the second round of review

In this second round of review, reviewer #3 has raised several minor points. We provide answers for the reviewer in black, changes to the manuscript text are shown in blue. To reply most effectively, we organized these points into the following main categories:

1- Study design

- In a few places, the reviewer asks questions regarding the TB status of household contacts (HHCs). To clarify, we did a longitudinal study in which cases can be any of the following: 1) recruited index TB cases, 2) identified co-prevalent TB patients in the household of index patients. These are cases among the HHCs who were identified as having TB within 14 days of enrollment of index TB cases, we refer to these as “baseline cases”, 3) identified incident TB events among HHCs followed for 12 months. We use the term “secondary cases” to identify HHCs who were diagnosed with TB after 14 days of enrollment of index cases. We refer to secondary cases whose *M. tb* strain shared exact MIRU genotyping with another TB case as “secondary clustered cases”. Controls are HHCs of index cases who were TST positive but who did not develop TB during 12 months of active follow-up. We have now added a section under results to clarify the case-control definition:

- Study design and case-control definition

We conducted a longitudinal, household contact genetic study of pulmonary TB in Lima, Peru (STAR Methods, Figures S1, and S2). All cases (N = 2105) have confirmed active TB. Within 14 days of enrollment of an index TB case (i.e. the first TB patient in each household), we screened their household contacts (HHCs) for signs and symptoms of active TB as well as for latent tuberculosis as measured by a tuberculin skin test (TST). These tests were repeated at 2, 6, and 12 months (STAR Methods, Figure 1). We refer to HHCs who were identified as having TB within 14 days of enrollment of index TB cases as “baseline cases”, and to HHCs who were diagnosed with TB after this period until the end of the 12 months follow up as “secondary” or “secondary clustered cases” (see STAR Methods for details). Controls (N = 1320) are HHCs of index cases who were TST positive but who did not develop TB during 12 months of active follow-up (STAR Methods). In addition to individuals' TB status, we also collected extensive information on sociodemographic risk factors for TB (STAR Methods, Table 1).

- In analyses focused on secondary or secondary clustered cases controls are those HHCs who live in the same households as secondary or secondary clustered cases, who were TST positive but who did not develop TB during 12 months of active follow-up. We have now added this specification to the methods section:

- In analyses focused on secondary or secondary clustered cases, controls were restricted to HHCs of these cases (i.e. if a household did not have any secondary or secondary clustered cases HHCs from that household were not included as controls).

- We did not exclude TB cases with a history of previous active TB. We now state this explicitly in the methods section.

- While cases were not excluded based on TB history, controls with a history of active TB or previous positive TST were excluded.

- In one analysis, described in lines 204-210, we selected TB cases and HHCs from different households. In this particular analysis, included TB cases (N=791) and HHCs (N=791) do not share their household (i.e. if a TB case was included no HHCs from the same household were included as a control for this analysis). Due to this selection, the sample size for this analysis is considerably smaller than our original analysis using the whole cohort (1,582 vs. 3,425). Despite the reduction in power, the effect of native Peruvian ancestry on TB progression risk is similar to our original analysis using the whole cohort with overlapping confidence intervals, suggesting that the two odds ratios are not statistically different (1.19 (1.10-1.28) vs 1.25 (1.18-1.33)).

- We appreciate the reviewer's suggestion about differences in our study's demographics compared to the general population, so we expanded the limitations section to underline that the demography of our cohort may be different from the general population.

- We note that the distribution of demographic variables such as sex, age, and education level in our cohort may differ from that in the general population. However, accounting for these covariates does not change our results, suggesting that our findings are unlikely to be driven by demographics.

2- Relatedness

- To clarify, only one analysis was done by removing the first and second-degree relatives. The result of that analysis is mentioned in line 162 and reported in detail in Figure S6 and Table S8. In all other analyses, all individuals (N = 3,425) were included; we corrected for relatedness by including a genetic relatedness matrix (GRM).

- We chose a kinship coefficient ≥ 0.125 , corresponding to second-degree relatives or closer, as our cutoff for relatedness in order to remove individuals with the highest degree of genetic relatedness without removing too many individuals and reducing our statistical power. Since the result of the analysis after removing the 430 related individuals (defined as above) remained the same as our initial analysis (Table S8), we did not perform further analyses by lowering the relatedness threshold and removing more distantly related individuals.

- The reviewer is correct to say that removing related individuals resulted in removing more HHCs than cases. As Figure S1 shows, when we selected HHCs for the genetic study, we tried to avoid selecting individuals who were first-degree relatives of an index TB case but did not exclude individuals if they were first-degree relatives of other HHCs. This resulted in a higher degree of relatedness among HHCs than between HHCs and TB cases.

3- Linearity of effects

In two places the reviewer raises questions about the possibility of a non-linear association between native Peruvian ancestry and/or self-reported race and TB progression risk.

- We note that the logistic regression model we used for our analyses is the standard practice for testing the association between covariates and case-control status¹. While non-linear associations between genetic ancestry and TB progression risk may exist, testing other models beyond the standard logistic regression falls beyond the scope of our manuscript.

- Based on our regression model a 10% increase in native Peruvian ancestry leads to a 25% increase in TB progression risk (odds ratio: 1.25, 95% CI=1.18-1.33). From this, the reviewer concludes that the top decile should have a 2.5-fold increase compared to the bottom decile. However, the expected 2.5-fold increase and the 3-fold increase that we get from directly comparing deciles (odds ratio: 2.90, 95% CI=1.99-4.26) are not significantly different as the confidence interval for these two effects are overlapping. That is confidence intervals for the top decile compared to the bottom decile based on the regression model can range from 1.8 to 3.3 fold with 2.5-fold being the average estimate.

- Subdividing the first decile of native Peruvian ancestry as a way of checking for a linear relationship between ancestry and TB progression risk is not a feasible approach; the average native Peruvian ancestry in our cohort is 80% meaning that there are relatively few individuals who have low levels of native Peruvian ancestry. Consequently, subdividing the first decile will result in high variance effect size estimates. The high variance plus the lower numbers in each decile will diminish our statistical power to detect any potential non-linear effects.

4- Admixture analysis

The reviewer raised concerns about the accuracy of our global ancestry inference and asked whether unsupervised clustering is an appropriate choice. We believe that our ancestry estimates using ADMIXTURE, which is widely used in the field are accurate for the reasons described below.

- ADMIXTURE algorithm is a priori unsupervised meaning it does not require reference populations with known ancestry. However, it has been previously shown that an unsupervised ADMIXTURE analysis that includes reference population samples performs similarly to supervised global ancestry inference methods^{2,3}.

- In line with the above point, we included reference populations from two separate datasets in our admixture analysis, the 1000 Genomes Project, and the Siberian and Native American populations from Reich et al, Nature 2012 (see Figures 3A and S5 for more details). Using these reference populations, we

ensure that appropriate reference populations that capture the genetic ancestry components of current Peruvians are included in our ADMIXTURE analysis.

- To further ensure that our unsupervised ADMIXTURE analysis performs similarly to supervised ancestry inference methods, we calculated global ancestry proportions for each individual from our local ancestry inference results by PCAdmix (a supervised ancestry inference method) and compared the results. Briefly, for each ancestry in each individual, global ancestry was calculated as the sum of local ancestry window lengths for that ancestry divided by the total length of all windows. As shown in the Response letter Figure 1 below, the results are highly concordant with our global ancestry results using ADMIXTURE.
- The inferred global ancestry proportions for Peruvians are in line with previous reports⁴⁻⁶.
- The information about the reference populations used in global ancestry analysis is provided in lines 159-163 of the manuscripts. Further details about the steps taken to merge and QC the data are provided in supplementary methods.
- The information about the reference populations used for supervised local ancestry analysis is provided in the methods section.
- The reviewer refers to the fact that European populations from the 1000 Genomes Project show some levels of admixture (Figure 3A) as an argument for the inaccuracy of ADMIXTURE analysis. We note that most modern-day populations have some levels of recent admixture that can be detected through genetic analysis. The populations included in the 1000 Genomes Project are no exception, and this has been shown in previous genetic studies of diverse populations⁵⁻⁷.

Response letter Figure 1: Correlation between global ancestry estimates using ADMIXTURE (K = 4, with the 1000 Genomes Project, and the Siberian and Native American populations from Reich et al, Nature 2012 as references) and local ancestry proportions calculated from local ancestry estimates using PCAdmix with four reference populations as described in the methods section. Each dot represents one individual (N = 3425). r = Pearson correlation coefficient. NAT: native Peruvian genetic ancestry, EUR: European genetic ancestry, AFR: West African genetic ancestry, ASI: East Asian genetic ancestry.

5- Using native American or indigenous ancestry instead of native Peruvian ancestry

- We prefer to continue to use the term native Peruvian ancestry when referring to the indigenous ancestry proportion of contemporary Peruvians. This decision was obtained after careful discussion with our Peruvian colleagues as well as Dr. David Jones, our co-author who is an expert in the history of science, questions related to science and race as well as race in the context of global and public health. We made this decision to avoid implying that our results can be generalized to all Native American

populations which can differ considerably in their genetics as well as in their culture and socio-demography.

- In line with the above point, at one point in the discussion, we mention that our study looks at “the effect of indigenous ancestry on TB or TB progression risk in South or Latin American populations” here we use the term indigenous ancestry to refer to the different native American ancestry components that exist among South and Latin American populations.

- To make the above points clear we have now modified the manuscript and defined native Peruvian genetic ancestry in the introduction:

- We then used genotype data to quantify the genetic diversity in our cohort and to estimate the proportion of native Peruvian genetic ancestry (i.e. the indigenous genetic ancestry component of the genome of contemporary Peruvians) in each individual.

6- Limitations and discussion of results

- We have now added a “Limitations of the Study” section to our manuscript as follows:

- Limitations of the study

One caveat of our study is that we have not tested for all possible non-genetic TB risk factors. For example, while we tried to account for factors related to exposure by including only participants with a documented household exposure to an index TB patient, we do not have information on the possible community or workplace exposures. We also could not correct for potential biases and unmeasured social discriminations and inequalities that might track with both genetic ancestry and TB progression risk. While these may be potential confounders, we consider it unlikely to explain the entirety of our signal: such biases are likely to track with households, and correction for household in our analyses did not alter our results. We note that the distribution of demographic variables such as sex, age, and education level in our cohort may differ from that in the general population. However, accounting for these covariates does not change our results suggesting that our findings are unlikely to be driven by specific demographics. Finally, we emphasize that while our study brings proof that TB risk can vary across populations with different genetic ancestries, our results cannot be generalized to all indigenous populations as different populations have different histories, and the sociodemographic factors, the primary determinants of TB risk, vary widely across different indigenous populations (Basta and de Sousa Viana, 2019; Cormier et al., 2019).

- Following the reviewer's suggestion we removed the word unbiased from the discussion and changed the corresponding sentence as follows:

- To our knowledge, this study is the first large-scale genetic study to look at the effect of indigenous ancestry on TB or TB progression risk in South or Latin American populations.

Here are a few other minor points that were brought up by reviewer #3, these points were already addressed in the manuscript or the previous response letter. Here we outline the line or figure numbers in which the answers are provided.

- We agree with the reviewer that self-reported race is correlated with genetic ancestry. We explicitly say that our results show such a correlation in lines 133-137.

- It has been said explicitly that we do not see an association between self-reported race and TB progression risk (lines 147-149).

- Following the previous round of revision, we have carefully reviewed our manuscript and removed the word match/matched/matching when referring to our study design. We did not refer to our study as a matched study design in the manuscript. The one occasion that reviewer #3 refers to the use of the word “matching” was taken from an answer to his/her previous comment and is not part of the manuscript.
- We stated in the discussion, explicitly, that the previously reported association on 3q23 cannot explain the association between native Peruvian ancestry and TB progression risk and explain the reasons for that conclusion in the discussion.
- We do not make any claim regarding the association of 5p23.2 with TB progression risk or any of the SNPs overlapping this locus (Table S13). In fact, we say explicitly that we did not find any local ancestry association in our admixture mapping analysis.
- Although the majority of our participants are adults, children were not systematically removed from our study. The age ranges are reported in Table 1.
- Nutritional status was obtained for everyone based on BMI, the WHO guidelines for BMI calculation are different for adults and children. To avoid confusion we modified the sentence about BMI in children as follows:
 - For children, we defined the nutritional status based on the World Health Organization BMI zscore tables (Onis et al., 2003).

Minor correction:

- We thank the reviewer for bringing up the mistake in the legend of figure S2B, and we fixed this error.

References:

1. Hoffman, J. I. E. Basic Biostatistics for Medical and Biomedical Practitioners (Second Edition). 581–589 (Academic Press, 2019).
2. Thornton, T. et al. Estimating and adjusting for ancestry admixture in statistical methods for relatedness inference, heritability estimation, and association testing. BMC Proc. 8, S5 (2014).
3. Thornton, T. A. & Bermejo, J. L. Local and global ancestry inference and applications to genetic association analysis for admixed populations. Genet. Epidemiol. 38 Suppl 1, S5–S12 (2014).
4. Norris, E. T. et al. Genetic ancestry, admixture and health determinants in Latin America. BMC Genomics 19, 861 (2018).
5. Homburger, J. R. et al. Genomic Insights into the Ancestry and Demographic History of South America. PLoS Genet. 11, e1005602 (2015).
6. Harris, D. N. et al. Evolutionary genomic dynamics of Peruvians before, during, and after the Inca Empire. Proc. Natl. Acad. Sci. U. S. A. 115, E6526–E6535 (2018).
7. Gurdasani, D. et al. The African Genome Variation Project shapes medical genetics in Africa. Nature 517, 327–332 (2015).